# Muscle-resident mesenchymal progenitors sense and repair peripheral nerve injury via the GDNF-BDNF axis

Kyusang Yoo[1], Young-Woo Jo[1], Takwon Yoo[1], Sang-Hyeon Hann[1], Inkuk Park[1], Yea-Eun Kim[1], Ye Lynne Kim[1], Joonwoo Rhee[1], In-Wook Song[1], Ji-Hoon Kim[1,2], Daehyun Baek[1], Young-Yun Kong[1]*

[1]School of Biological Sciences, Seoul National University, Seoul, Republic of Korea; [2]Molecular Recognition Research Center, Korea Institute of Science and Technology, Seoul, Republic of Korea

*For correspondence: ykong@snu.ac.kr

Competing interest: The authors declare that no competing interests exist.

**Abstract** Fibro-adipogenic progenitors (FAPs) are muscle-resident mesenchymal progenitors that can contribute to muscle tissue homeostasis and regeneration, as well as postnatal maturation and lifelong maintenance of the neuromuscular system. Recently, traumatic injury to the peripheral nerve was shown to activate FAPs, suggesting that FAPs can respond to nerve injury. However, questions of how FAPs can sense the anatomically distant peripheral nerve injury and whether FAPs can directly contribute to nerve regeneration remained unanswered. Here, utilizing single-cell transcriptomics and mouse models, we discovered that a subset of FAPs expressing GDNF receptors *Ret* and *Gfra1* can respond to peripheral nerve injury by sensing GDNF secreted by Schwann cells. Upon GDNF sensing, this subset becomes activated and expresses *Bdnf*. FAP-specific inactivation of *Bdnf* (*Prrx1*^Cre; *Bdnf*^fl/fl) resulted in delayed nerve regeneration owing to defective remyelination, indicating that GDNF-sensing FAPs play an important role in the remyelination process during peripheral nerve regeneration. In aged mice, significantly reduced *Bdnf* expression in FAPs was observed upon nerve injury, suggesting the clinical relevance of FAP-derived BDNF in the age-related delays in nerve regeneration. Collectively, our study revealed the previously unidentified role of FAPs in peripheral nerve regeneration, and the molecular mechanism behind FAPs' response to peripheral nerve injury.

## eLife assessment

The study has identified a cell type in muscle that is characterized as an adipogenic progenitor cell that is capable of promoting regeneration through the action of BDNF, a prominent growth factor regulated by GDNF in Schwann cells. These results represent an **important** cellular explanation for nerve regeneration. The revised analysis is **solid** but the work remains **incomplete** due to a lack of evidence that BDNF is produced during the process through the action of GDNF.

## Introduction

Positioned in the interstitial space between myofibers (*Uezumi et al., 2010*), fibro-adipogenic progenitors (FAPs) interact with cellular components within a skeletal muscle to ensure normal development, homeostasis, and regeneration of muscle tissue. During developmental myogenesis, embryonic FAPs expressing *Osr1* contribute to limb muscle patterning by regulating the expression of extracellular matrix (ECM) genes that makeup muscle connective tissue (*Vallecillo-García et al., 2017*). In young adults, FAPs are necessary for normal growth and long-term maintenance of skeletal muscle, which otherwise undergo progressive muscle atrophy in the absence of PDGFRα⁺ FAPs (*Roberts et al.,*

*2013*; *Wosczyna et al., 2019*; *Uezumi et al., 2021*). Upon muscle injury, FAPs proliferate in response to IL-4/IL-13 signals from eosinophils, and participate in the clearing of necrotic debris via phagocytosis (*Heredia et al., 2013*). Also, the proliferated FAPs regulate the expansion and asymmetric commitment of muscle stem cells (MuSCs) via secreted factors such as WISP1, leading to robust de novo myofiber formation (*Joe et al., 2010*; *Lukjanenko et al., 2019*; *Wosczyna et al., 2019*). Conversely, the absence of FAPs or its functional decline with age cause premature differentiation of MuSCs upon injury, resulting in the formation of smaller regenerated myofibers (*Murphy et al., 2011*; *Lukjanenko et al., 2019*; *Wosczyna et al., 2019*). After sufficient regeneration of myofibers occurs, FAPs undergo apoptosis via TNF signaling from monocyte/macrophages, such that its numbers return to those of unperturbed muscles (*Lemos et al., 2015*; *Saito et al., 2020*). Failure to remove excess FAPs after muscle regeneration results in unwanted fibrosis, which compromises muscle function (*Uezumi et al., 2011*; *Uezumi et al., 2014*).

In addition to the formation and maintenance of muscle tissue, FAPs also contribute to the maturation and maintenance of the neural components within skeletal muscle. Previously, we reported that FAPs promote postsynaptic maturation of the neuromuscular junction (NMJ) through the BAP1/SMN axis during postnatal development (*Kim et al., 2022*). Selective inactivation of *Bap1* in FAPs results in dysfunctional NMJs, with sustained expression of the immature form of acetylcholine receptor subunit, *AchRγ*, in skeletal muscle (*Kim et al., 2022*). Progressively, these mice exhibit denervation at the NMJ, retraction of motor axons, reduction of myelination and axon diameter, and eventually motor neuron loss, suggesting that FAPs prevent the dying-back loss of motor neurons (*Kim et al., 2022*). Recently, we also reported defective presynaptic maturation and maintenance in mice with selective *Smn* downregulation in FAPs, again suggesting the role of FAPs in postnatal NMJ development (*Hann et al., 2024*). In adult mice, BMP3B secretion by FAPs stabilize NMJs and Schwann cells by promoting the myelination program in Schwann cells, thereby directly contributing to the maintenance of neural components within skeletal muscle (*Uezumi et al., 2021*). In the absence of FAPs or *Bmp3b*, mice exhibit muscle weakness and myofiber atrophy along with destabilization of Schwann cells and denervation at NMJs, which closely resemble the phenotypes observed in age-related sarcopenia (*Uezumi et al., 2021*). Similarly, conditional deletion of *Bap1* in FAPs in adulthood cause denervation at the NMJs and eventually loss of motor neurons, demonstrating the requirement of FAPs in maintaining the neuromuscular system (*Kim et al., 2022*). Conversely, disturbance in the neural component can influence the behavior of FAPs; for instance, denervation is known to activate FAPs (*Contreras et al., 2016*; *Gonzalez et al., 2017*; *Madaro et al., 2018*). This suggests that FAPs can somehow sense the anatomically distant peripheral nerve injury. However, the question of how FAPs are able to sense the distant peripheral nerve injury remains unanswered (*Theret et al., 2021*). Furthermore, whether FAPs are actually able to exert beneficial effects on peripheral nerve regeneration remains elusive.

In accordance with its various functions, heterogeneity within FAPs began to be recognized with the advent of single-cell analysis technology (*Contreras et al., 2021*). By profiling the expression levels of 87-selected genes in isolated singlets of FAPs, dynamic transitions between heterogeneous subpopulations of FAPs, identified by different expression levels of TIE2 and VCAM1, was observed during postnatal and regenerative myogenesis (*Malecova et al., 2018*). The report showed that while activation of TIE2$^{high}$ FAPs is observed in neonatal mice, activation of VCAM1$^+$ FAPs is observed in injured muscles, suggesting distinct functional involvement of FAPs in the two different contexts of myogenesis (*Malecova et al., 2018*). Additionally, single-cell RNA-sequencing (scRNA-seq) enabled the identification of heterogeneity within FAPs based on the genome-wide transcriptome data (*Lieberman et al., 2021*). In homeostatic adult muscle, two distinct subpopulations within FAPs have been identified, namely *Dpp4*$^+$ and *Cxcl14*$^+$ FAPs (*Scott et al., 2019*; *Oprescu et al., 2020*). Functionally, we reported that DPP4$^+$ FAPs contribute to the maturation and maintenance of the neuromuscular system via the BAP1/SMN axis (*Kim et al., 2022*). In juvenile muscle, five different subpopulations within FAPs were characterized, each having different contexts of activation and differentiation potentials (*Leinroth et al., 2022*). *Osr1*$^+$ FAPs are precursor cells that can form all other subpopulations; *Clu*$^+$ FAPs are most potent in mineralization; *Adam12*$^+$ and *Gap43*$^+$ FAPs are immune-responsive; and *Hsd11b1*$^+$ FAPs respond to nerve transection (*Leinroth et al., 2022*). The different activation cues and differentiation potentials in each subpopulation of FAPs suggest distinct roles those subsets can play in different contexts of skeletal muscle biology. Indeed, dynamic transcriptomic changes in FAP subpopulations in response to muscle injury (*Scott et al., 2019*; *De Micheli et al., 2020*; *Oprescu*

*et al., 2020*) or denervation *Nicoletti et al., 2020*; *Proietti et al., 2021*; *Lin et al., 2022*; *Nicoletti et al., 2023* have been identified in studies that implemented scRNA-seq analysis. Still, the question of how FAPs may sense the distant nerve injury and whether FAPs can beneficially contribute to nerve regeneration remain largely unknown.

In response to nerve injury, various neurotrophic factors are expressed and secreted by the surrounding cells to facilitate regeneration. One of those neurotrophic factors is GDNF, previously reported to be expressed robustly by Schwann cells upon nerve injury (*Hammarberg et al., 1996*; *Höke et al., 2002*; *Arthur-Farraj et al., 2012*; *Xu et al., 2013*; *Proietti et al., 2021*). Canonical GDNF signaling pathway involves two well-known receptors of GDNF: RET and GFRα1. Binding of GDNF to GFRα1 induces complex formation with the receptor tyrosine kinase (RTK) RET, which initiates the downstream phosphorylation cascade via autophosphorylation upon dimerization (*Jing et al., 1996*; *Treanor et al., 1996*; *Trupp et al., 1996*). The downstream phosphorylation cascades include the Ras-MAPK pathway, the PI3K-Akt pathway, and the Src family kinase-mediated pathway, which are known to promote neuronal survival and neurite outgrowth (*Encinas et al., 2001*; *Sariola and Saarma, 2003*). Indeed, exogenous delivery of GDNF was shown to promote motor neuron survival and enhance axonal growth upon nerve injury, which resulted in improved functional recovery (*Cintrón-Colón et al., 2020*). Though the role and function of GDNF on the regenerating neuron have been demonstrated, other possible cellular targets of GDNF that may facilitate the nerve regeneration process remain to be studied.

Here, using the scRNA-seq approach, we aimed to identify the response mechanism of FAPs to nerve injury, by uncovering its nerve injury-sensing mechanism and its potentially beneficial effect on nerve regeneration. To obtain a comprehensive scRNA-seq database of FAPs' response to nerve injury, FAPs from both chronic, non-regenerating nerve injury (denervation)- and acute, regeneration-prone nerve injury (crush)-affected muscles were collected at different time points over the course of regeneration. As a result, distinct transcriptomic profiles of FAPs at different time points post the two types of nerve injuries were captured in single-cell resolution, from which the response mechanism of FAPs to nerve injury was identified and validated using mouse models. Specifically, we found that upon peripheral nerve injury, GDNF from Schwann cells can activate FAPs, which in turn express BDNF to promote remyelination during nerve regeneration. Our study suggests FAPs as an important player that actively participates in the nerve regeneration process, of which we believe should be considered in future studies aiming for an improved understanding of the peripheral nerve regeneration process.

## Results
### Single-cell transcriptome profiling of nerve injury-affected FAPs

To establish a comprehensive transcriptome database of nerve injury-affected FAPs at single-cell resolution, we performed scRNA-seq using FAPs isolated from sciatic nerve crush injury (SNC)- or denervation (DEN)-affected muscles at different time points over the course of regeneration (*Figure 1A*, *Figure 1—figure supplement 1*). Four-time points (3, 7, 14, and 28 days post-injury, hereafter dpi) along the regeneration process were chosen for analysis to capture the transcriptomes of FAPs at both early and late stages of regeneration. Selection of such time points was based on a previous report that showed the reinnervation process of the tibialis anterior (TA) muscle after SNC, where Wallerian degeneration was evident at 7 dpi, and reinnervation was mostly completed at 28 dpi (*Magill et al., 2007*). Including the uninjured control, scRNA-seq data from a total of nine samples (Uninjured, SNC-3dpi, SNC-7dpi, SNC-14dpi, SNC-28dpi, DEN-3dpi, DEN-7dpi, DEN-14dpi, and DEN-28dpi) were obtained via the 10x Genomics platform (*Figure 1A*). Quality control and filtering of the sequenced cells yielded a total of 44,597 cells for further analysis, where 4955.2±1022.8 cells were captured from each sample. Prior to downstream analysis, integration (*Hao et al., 2021*) of our scRNA-seq data with a publicly available scRNA-seq data of mononuclear cells from denervated muscles at 0, 2, 5, and 15 dpi (*Nicoletti et al., 2023*) was carried out for data validation. Expectedly, most (97.7%) of the filtered cells in our scRNA-seq data clustered with the denervation-affected FAPs in the published scRNA-seq data, confirming the validity of the data produced in our study (*Figure 1—figure supplement 2*).

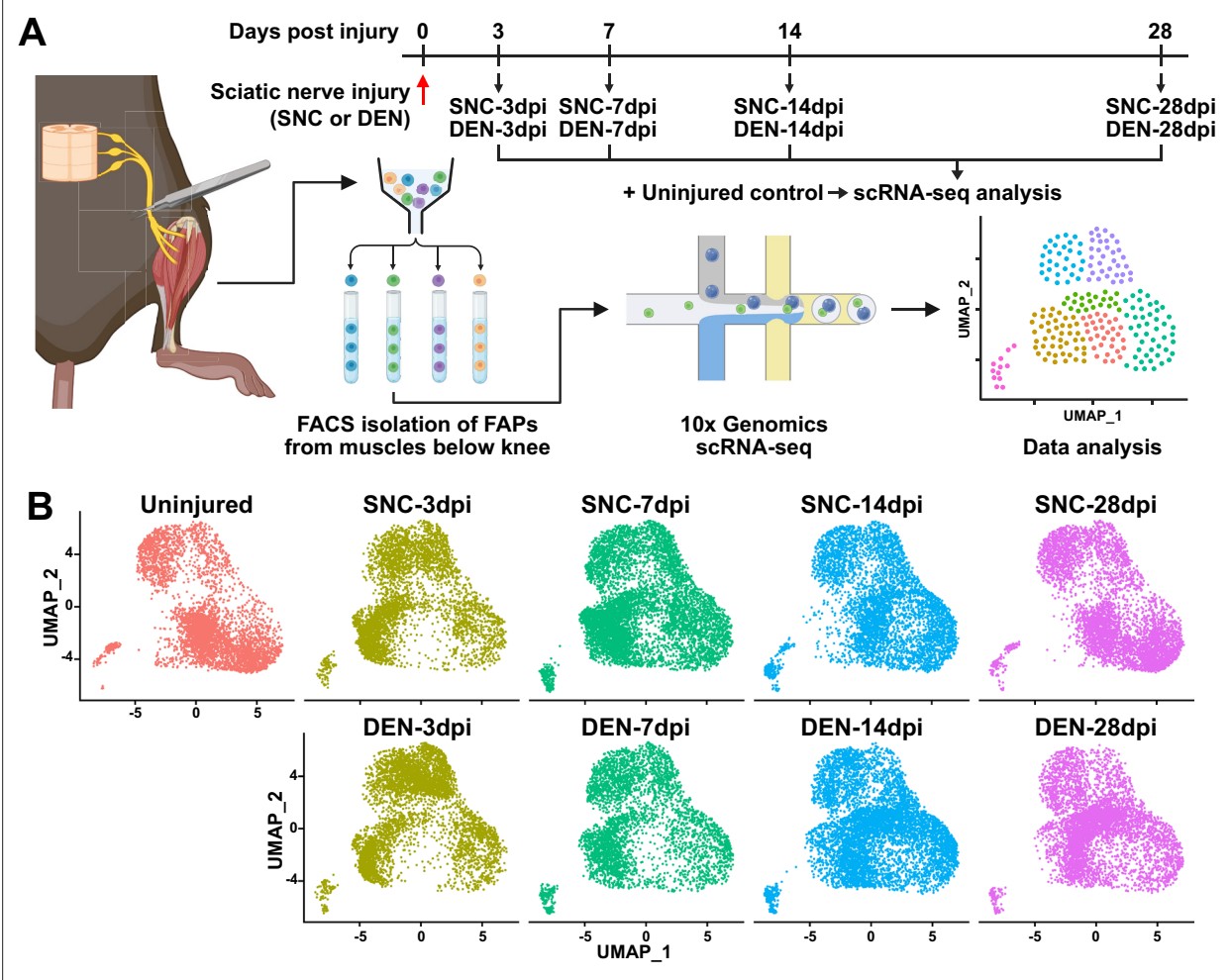

**Figure 1.** Single-cell transcriptome profiling of nerve injury-affected fibro-adipogenic progenitors (FAPs). (**A**) Experimental scheme depicting the procedures for sample collection and single-cell RNA-sequencing (scRNA-seq). The types of nerve injuries and time points for FAP isolation for each sample are specified. (**B**) Single-cell transcriptome data of nerve injury-affected FAPs displayed separately by samples on uniform manifold approximation and projection (UMAP) plots.

The online version of this article includes the following figure supplement(s) for figure 1:

**Figure supplement 1.** Fluorescence-activated cell sorting (FACS) isolation of muscle-resident fibro-adipogenic progenitors (FAPs).

**Figure supplement 2.** Validation of single-cell RNA-sequencing (scRNA-seq) data produced in this study.

## Distinct response profiles of FAPs upon nerve crush injury versus denervation

To look into the chronological transcriptomic changes that occur in FAPs in response to SNC or DEN on a global level, we analyzed our scRNA-seq data by samples. Visualization of the scRNA-seq data on uniform manifold approximation and projection (UMAP) plots showed similar changes in the early stages of regeneration (3 and 7 dpi) compared to uninjured FAPs, regardless of the type of injury (*Figure 1B*). However, as FAPs reached later stages of regeneration (14 and 28 dpi), SNC-affected FAPs returned to states similar to uninjured control, while DEN-affected FAPs stayed in the activated state (*Figure 1B*). The similarities and differences observed on the UMAP plots could also be found in the differentially expressed gene (DEG) analyses as well as in the hierarchical clustering analysis using those DEGs (*Figure 2A–D*, *Figure 2—figure supplement 1*). As a result of pairwise DEG analyses comparing all nine samples, different numbers of DEGs were identified, of which correlated with the

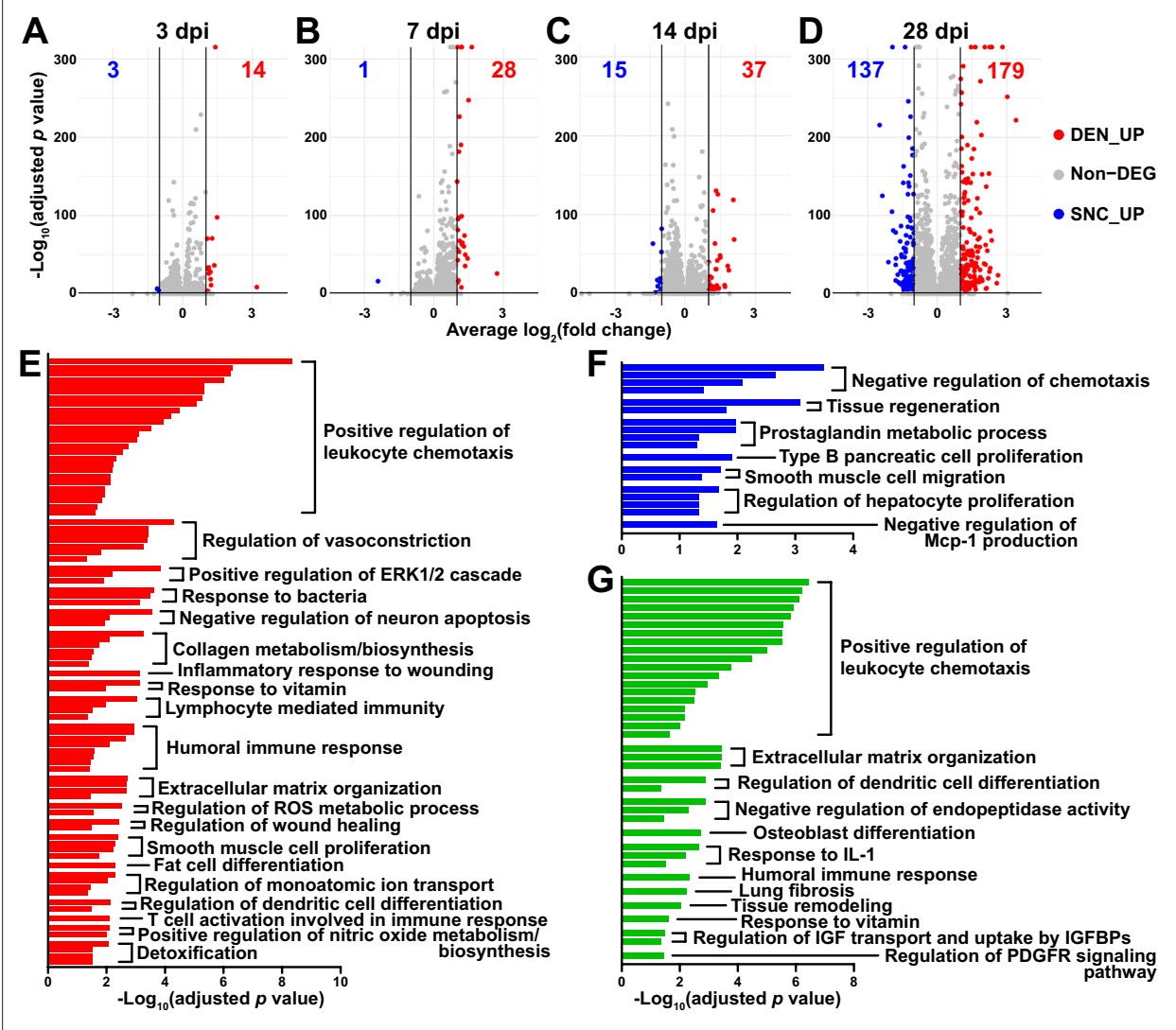

**Figure 2.** Distinct response profiles of fibro-adipogenic progenitors (FAPs) upon nerve crush injury versus denervation. (**A–D**) Volcano plots showing different numbers of differentially expressed genes (DEGs) identified from comparing sciatic nerve crush (SNC)- versus denervation (DEN)-affected FAPs at (**A**) 3, (**B**) 7, (**C**) 14, (**D**) 28 days post injury (dpi). (**E–G**) Pathway terms enriched from gene set overrepresentation analyses using g:Profiler. DEGs used as input were (**E**) DEN-28dpi-upregulated versus SNC-28dpi, (**F**) SNC-28dpi-upregulated versus DEN-28dpi, and (**G**) DEGs upregulated commonly in SNC-3dpi, SNC-7dpi, DEN-3dpi, and DEN-7dpi versus uninjured control.

The online version of this article includes the following figure supplement(s) for figure 2:

**Figure supplement 1.** Differentially expressed gene (DEG) analyses reveal similarities and differences between fibro-adipogenic progenitors (FAPs) affected by sciatic nerve crush (SNC) or denervation (DEN) at different time points.

**Figure supplement 2.** Expression patterns of *Il6* and *Stat3* in nerve injury-affected fibro-adipogenic progenitors (FAPs).

**Figure supplement 3.** Gene set overrepresentation analyses using differentially expressed genes (DEGs) from pairwise comparisons of the nine single-cell RNA-sequencing (scRNA-seq) samples.

similarities between samples observed on UMAP plots (*Figure 1B*, *Figure 2A–D*, *Figure 2—figure supplement 1A ,B*). Hierarchical clustering of the nine samples using all unique DEGs identified from the pairwise comparisons showed clustering of uninjured control with SNC-14dpi and SNC-28dpi, suggesting FAPs' return to homeostatic state (*Figure 2—figure supplement 1C*). On the other hand, DEN-14dpi and DEN-28dpi clustered with each other, but not with uninjured control, suggesting the chronic activation of FAPs in response to DEN as reported previously (*Madaro et al., 2018*; *Figure 2—figure supplement 1C*). Samples that captured the early responses of FAPs to nerve injuries (3 or 7 dpi) were clustered together by dpi rather than the type of injury, suggesting similar response profiles

of FAPs to both types of nerve injuries (*Figure 2—figure supplement 1C*). Indeed, the numbers of DEGs between SNC-3dpi versus DEN-3dpi and SNC-7dpi versus DEN-7dpi were among the lowest identified in the pairwise analyses (*Figure 2A and B*, *Figure 2—figure supplement 1A*). Overall, the number of DEGs between SNC- and DEN-affected FAPs increased significantly with dpi, showing the bifurcation of FAP's response to the different types of nerve injuries in the later stages of regeneration (*Figure 2A–D*).

In a previous study, chronic activation of the STAT3/IL-6 pathway in FAPs in response to DEN was reported (*Madaro et al., 2018*). Indeed, *Il6* was identified as one of the genes upregulated at all four-time points in response to DEN compared to uninjured control in our data (*Figure 2—figure supplement 2A, B*). Although it did not pass the fold change (FC) threshold (FC ≥ 2) in the DEG analyses, expression of *Stat3* also showed significant chronic upregulation in all DEN-affected FAPs as well (*Figure 2—figure supplement 2C, D*). In response to SNC, however, only transient upregulation of both genes was observed (*Figure 2—figure supplement 2*).

To obtain biological insights on the different responses of FAPs to SNC versus DEN in the later stage of regeneration, we subjected the two lists of genes from *Figure 2D* (28 dpi, SNC_UP and DEN_UP) to gene set overrepresentation analysis (ORA) using g: Profiler (*Kolberg et al., 2023*). From both sets of genes, pathways related to tissue regeneration/wound healing were enriched (*Figure 2E and F*, *Figure 2—figure supplement 3A, B*). In contrast, pathways related to immune cell recruitment, inflammation, and ECM regulation by collagen biosynthesis were enriched specifically in DEN-28dpi (*Figure 2E*, *Figure 2—figure supplement 3A*), which is consistent with the previous report that showed the direct contribution of FAPs to fibrosis in denervated muscles (*Contreras et al., 2016*; *Madaro et al., 2018*). Also, mild immune cell infiltration into affected muscles in response to DEN was previously described (*Lin et al., 2022*; *Nicoletti et al., 2023*); our results suggest the role of FAPs in chemotactic recruitment of immune cells in denervated muscles. In addition, the pathway 'negative regulation of neuron apoptosis' was enriched in DEN-28dpi, suggesting a prolonged attempt of FAPs to preserve neurons that must be alive for reinnervation of the denervated muscle (*Figure 2E*, *Figure 2—figure supplement 3A*). On the other hand, some pathways were exclusively enriched in SNC-28dpi, such as negative regulation of chemotaxis and prostaglandin metabolism (*Figure 2F*, *Figure 2—figure supplement 3B*). It is generally understood that after a successful tissue regeneration process, the resolution of the immune response allows for the tissue to return to homeostasis (*Ortega-Gómez et al., 2013*; *Aurora and Olson, 2014*; *Julier et al., 2017*); our results suggest the role of FAPs in regulating immune resolution near the end of nerve regeneration. Furthermore, the role of prostaglandin in peripheral nerve regeneration has recently been described (*Forese et al., 2020*; *Bakooshli et al., 2023*); our ORA results suggested that FAPs may also be involved in the regulation of prostaglandin levels during peripheral nerve regeneration.

To discover biological pathways behind the supposedly similar responses of FAPs to both SNC and DEN in the early phases of regeneration, we examined DEGs by comparing SNC-3dpi, SNC-7dpi, DEN-3dpi, and DEN-7dpi to uninjured controls. Many of the upregulated genes identified in the DEG analyses were shared amongst the four samples (*Figure 2—figure supplement 3C*). ORA using the shared upregulated genes revealed enrichment in pathways that were also enriched in DEN-28dpi compared to SNC-28dpi, such as immune cell recruitment and ECM regulation, supporting the idea that early-activated states within FAPs persist for a prolonged period in response to DEN (*Figure 2E and G*, *Figure 2—figure supplement 3A and D*). Collectively, analysis of our scRNA-seq data by samples revealed similar response profiles of FAPs to both SNC and DEN in the early stages of regeneration, which then bifurcated into chronic activation in response to DEN and return to homeostasis in response to SNC, showing correlative behaviors along with the degree of nerve regeneration and target muscle reinnervation.

## Nerve injury-responsive subsets within FAPs

Although analysis of our scRNA-seq data on the population level provided general insights on how FAPs may respond to the different types of nerve injuries, the results could not provide us with sufficient clues on how FAPs can sense nerve injuries, or how they may directly contribute to nerve regeneration. Thus, we next analyzed our scRNA-seq data on the subpopulation level, hoping to distinguish subsets within FAPs that may be more relevant to the context of sensing and responding to nerve injury. To identify distinct subsets within nerve injury-affected FAPs, we applied unsupervised

clustering to the merged scRNA-seq data of all nine samples following the Seurat-R workflow (*Hao et al., 2021*). As a result, seven clusters with unique gene expression profiles were identified from the nerve injury-affected FAPs, with marker genes specifically expressed in each cluster (*Figure 3A–C*, *Figure 3—figure supplement 1*).

Interestingly, while clusters 4–7 showed little or no significant change in their proportions in response to nerve injury, clusters 1–3 exhibited dramatic changes upon nerve injury (*Figure 3D and E*). In particular, cluster 1 was mostly present in uninjured muscles or in muscles where reinnervation had occurred to at least some degree (SNC-14dpi and SNC-28dpi) (*Magill et al., 2007*; *Figure 3D and E*). In contrast, the presence of clusters 2 and 3 were mutually exclusive to cluster 1, such that their appearances were transient in response to SNC and chronic upon DEN (*Figure 3D and E*). Based on this mutual exclusivity of the three clusters, we speculated that cluster 1 can sense and respond to nerve injuries, and that clusters 2 and 3 may have arisen from cluster 1 upon nerve injury.

To obtain clues on whether such changes between FAP clusters could have actually occurred in response to nerve injury, we first performed RNA velocity analysis using R package velocyto.R (*La Manno et al., 2018*). RNA velocities on the UMAP plots predicted transcriptomic flow from cluster 1 to clusters 2 and 3 in the early stages of regeneration in both SNC- and DEN-affected FAPs, which was in line with our speculation (*Figure 3—figure supplement 2*). Conversely, transcriptomic flow from clusters 2 and 3 back to cluster 1 was evident in SNC-affected FAPs in the later stages of regeneration (*Figure 3—figure supplement 2*). However, RNA velocities in DEN-affected FAPs were represented as dots instead of arrows on clusters 2 and 3 in the later stages, suggesting an unchanging, chronic state of their transcriptomes, which is consistent with the chronic activation of FAPs in response to DEN (*Madaro et al., 2018*; *Figure 3—figure supplement 2*). Additionally, hierarchical clustering of the seven FAP clusters using DEGs enriched in each cluster grouped clusters 1–3 together, supporting our speculation that clusters 2 and 3 originate from cluster 1 (*Figure 3—figure supplement 3*). Recently, *Hsd11b1*-expressing FAPs were identified as the FAP subset that is specifically activated in response to nerve transection injury (*Leinroth et al., 2022*). Since we speculated that cluster 1 in our scRNA-seq data can sense and respond to nerve injury, we examined the expressions of marker genes identified in the previous report – *Hsd11b1*, *Mme*, *Ret*, and *Gfra1* (*Leinroth et al., 2022*) – in our scRNA-seq data. Indeed, all four markers were enriched in cluster 1 in our data (*Figure 3—figure supplement 4*). Expressions of marker genes *Hsd11b1* and *Mme* were also enriched in clusters 2 and 3, further supporting the idea that those clusters may have arisen from cluster 1 (*Figure 3—figure supplement 4A–C*). Together, these data suggest that clusters 1–3 are the dynamic interchanging subsets of FAPs that specifically respond to nerve injury, where cluster 1 senses nerve injuries in unperturbed muscles and clusters 2 and 3 arise from cluster 1 to respond to nerve injury.

## GDNF signaling pathway in the nerve injury-sensing mechanism by FAPs

Among the four marker genes expressed in cluster 1, *Ret* and *Gfra1* are well-known as GDNF receptors, where GFRα1 directly binds GDNF, which in turn activates the RTK RET for downstream signal transduction (*Jing et al., 1996*; *Treanor et al., 1996*; *Trupp et al., 1996*). Meanwhile, robust but specific expression of GDNF by Schwann cells in response to peripheral nerve injury has been reported (*Hammarberg et al., 1996*; *Höke et al., 2002*; *Arthur-Farraj et al., 2012*; *Xu et al., 2013*; *Proietti et al., 2021*). Accordingly, we presumed that FAPs, especially the *Ret*- and *Gfra1*-expressing cluster 1 cells, may sense the distant nerve injury by detecting GDNF secreted from Schwann cells. Notably, both *Ret* and *Gfra1* were among the top 10 DEGs specifically enriched in cluster 1, suggesting that they can readily respond to GDNF (*Figure 4A*). In addition, comparing the expression levels of *Ret* and *Gfra1* in skeletal muscle-resident mononuclear cell populations isolated by fluorescence-activated cell sorting (FACS) revealed robust co-expression of both genes in FAPs, but not as much in other cell populations (*Figure 4B and C*). Thus, FAPs may be the main cell type within skeletal muscle that can respond to GDNF secreted by Schwann cells in case of a nerve injury.

To further investigate the relevance of GDNF signaling in the nerve injury-sensing mechanism by FAPs, we subjected lists of DEGs enriched in clusters 1–3 to ORA. Genes enriched in cluster 1 returned pathways 'GDNF receptor signaling pathway,' 'regulation of cellular response to growth factor stimulus,' and 'positive regulation of peptidyl-tyrosine phosphorylation,' where tyrosine residues on the RTK RET is known to be phosphorylated upon activation (*Jing et al., 1996*; *Treanor et al., 1996*;

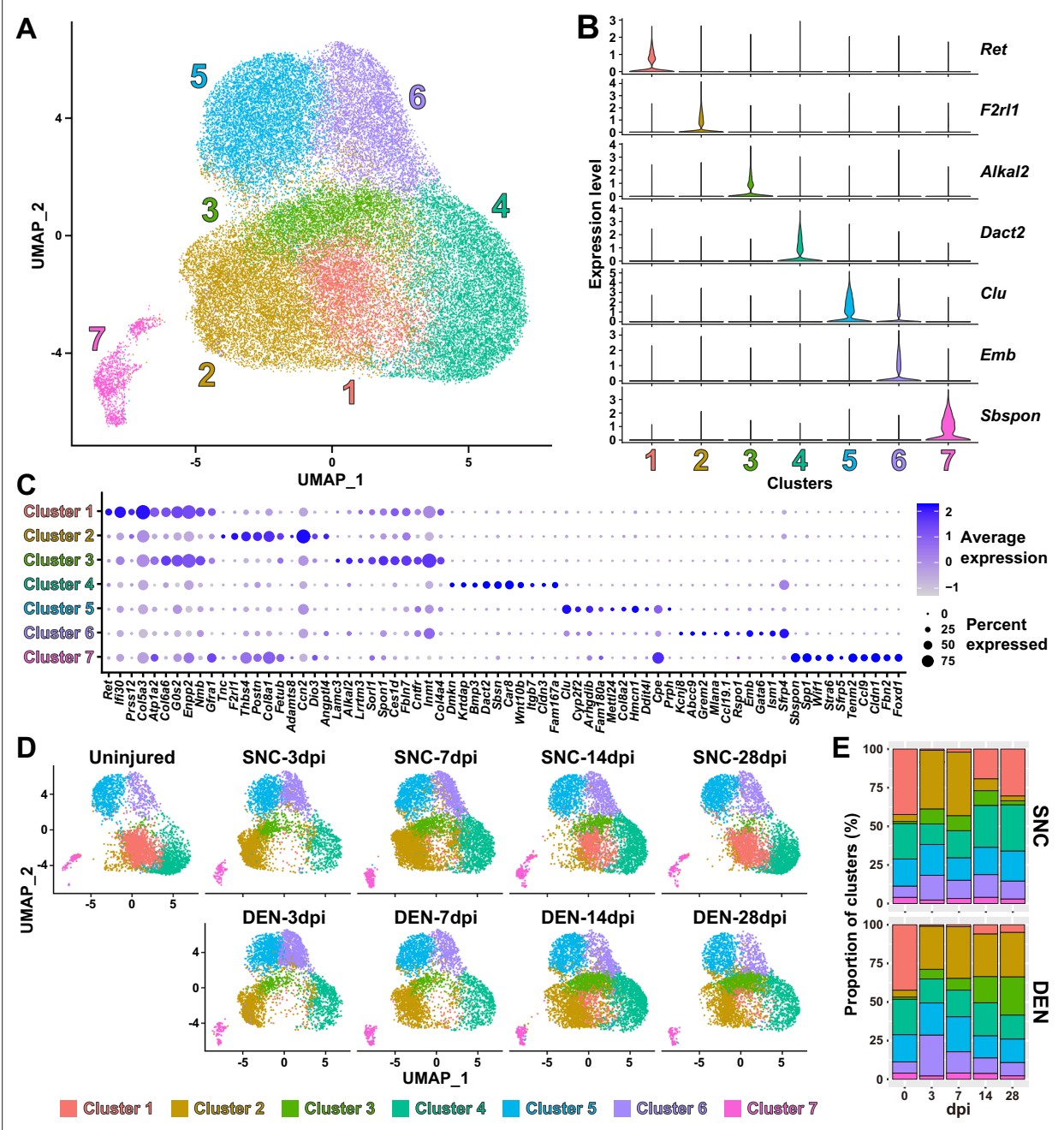

**Figure 3.** Nerve injury-responsive subsets within fibro-adipogenic progenitors (FAPs). (**A**) Seven clusters were identified by unsupervised clustering using all nine single-cell RNA-sequencing (scRNA-seq) samples obtained in this study displayed on the uniform manifold approximation and projection (UMAP) plot. (**B**) Violin plots showing expressions of unique marker genes identified in each cluster. (**C**) Dotplot showing the expression levels and percentages of the top 10 differentially expressed genes (DEGs) enriched in each cluster. (**D**) UMAP plots of clustered scRNA-seq data displayed separately by samples. (**E**) Barplots showing the proportions of the seven clusters that comprise each scRNA-seq sample of nerve injury-affected FAPs. For 0 dpi, data from the same uninjured control sample is displayed for both SNC and DEN.

The online version of this article includes the following figure supplement(s) for figure 3:

**Figure supplement 1.** Marker gene expression in each fibro-adipogenic progenitor (FAP) cluster.

**Figure supplement 2.** Transcriptomic flow between fibro-adipogenic progenitor (FAP) clusters.

**Figure supplement 3.** Relatedness between fibro-adipogenic progenitor (FAP) clusters.

**Figure supplement 4.** Expression of nerve transection-responsive fibro-adipogenic progenitor (FAP) subset-specific genes reported by *Leinroth et al., 2022* in the seven FAP clusters identified in this study.

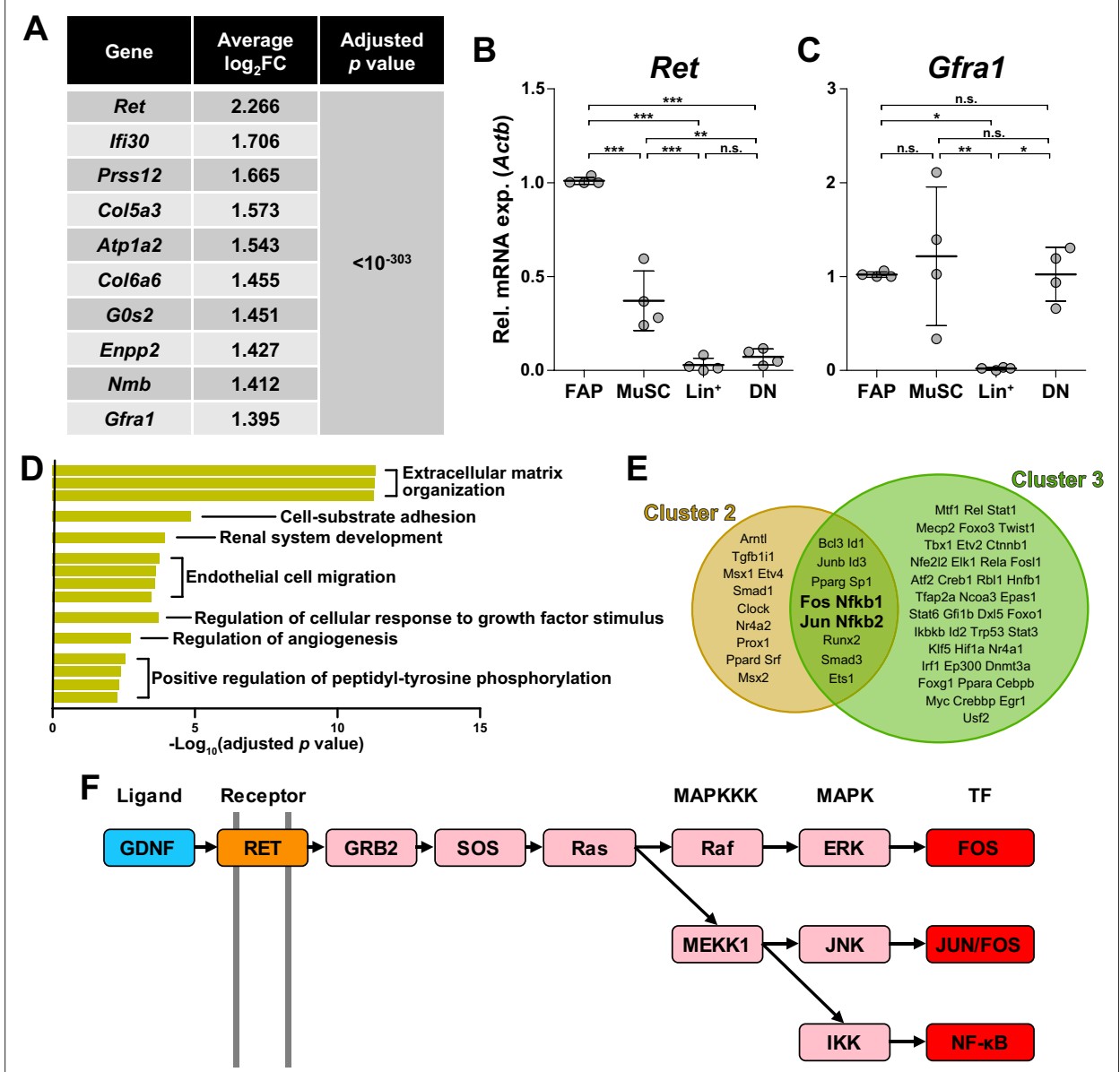

**Figure 4.** Glial cell line-derived neurotrophic factor (GDNF) signaling pathway in the nerve injury-sensing mechanism by fibro-adipogenic progenitors (FAPs). (**A**) Top 10 genes specifically enriched in cluster 1 FAPs. p-values were drawn from the Wilcoxon rank sum test. (**B, C**) RT-qPCR results show the expressions of (**B**) *Ret* and (**C**) *Gfra1* in mononuclear cells isolated from uninjured muscles by fluorescence-activated cell sorting (FACS). MuSC, muscle stem cells; Lin+, lineage-positive cells; DN, Vcam1/Sca1 double-negative cells. n=4; one-way ANOVA with Bonferroni's post hoc test. *p<0.05, **p<0.01, ***p<0.001, n.s., not significant. (**D**) Shared pathway terms commonly identified from gene set overrepresentation analyses using differentially expressed genes (DEGs) specifically upregulated in clusters 1, 2, or 3. See *Figure 4—figure supplements 1–3* for the full overrepresentation analysis (ORA) results. (**E**) Venn diagram showing the results from TRRUST analyses using DEGs enriched in clusters 2 and 3. Transcription factors predicted to regulate genes upregulated in each cluster are listed. (**F**) Simplified diagram of the GDNF/RET-MAPK signaling pathway. Blue: GDNF ligand; orange: GDNF receptor RET expressed in cluster 1; pink: downstream cascade genes expressed in clusters 1–3; red: transcription factors commonly predicted to regulate upregulated genes in clusters 2 and 3.

The online version of this article includes the following figure supplement(s) for figure 4:

**Figure supplement 1.** Pathway terms enriched in cluster 1 fibro-adipogenic progenitors (FAPs).

**Figure supplement 2.** Pathway terms enriched in cluster 2 fibro-adipogenic progenitors (FAPs).

**Figure supplement 3.** Pathway terms enriched in cluster 3 fibro-adipogenic progenitors(FAPs).

**Figure supplement 4.** Glial cell line-derived neurotrophic factor (GDNF) signaling pathway within the MAPK signaling pathway.

*Trupp et al., 1996*; *Figure 4—figure supplement 1*). The two latter pathways were also found in the ORA results using DEGs enriched in clusters 2 and 3, supporting the idea that those clusters originate from cluster 1 upon nerve injury (*Figure 4D*, *Figure 4—figure supplements 1–3*). The pathway 'positive regulation of ERK1/2 cascade' was enriched in cluster 2, suggesting the involvement of GDNF-RET-Ras-ERK signaling cascade within the MAPK signaling pathway in this cluster, which is one of the known downstream pathways of canonical GDNF signaling (*Airaksinen and Saarma, 2002*; *Sariola and Saarma, 2003*; *Kanehisa et al., 2023*; *Figure 4—figure supplements 2 and 4*). In addition to ORA, we predicted upstream transcription factors (TFs) that could have regulated the expressions of the DEGs enriched in the two activated FAP subsets, clusters 2 and 3, using TRRUST (*Han et al., 2018*). As a result, TFs *Fos*, *Jun* and NF-κB (*Nfkb1, Nfkb2*) were predicted from both lists of DEGs, all of which are known to act downstream of the GDNF/RET-induced MAPK signaling pathway (*Fielder et al., 2018*; *Kanehisa et al., 2023*; *Figure 4E and F*, *Figure 4—figure supplement 4*, *Supplementary file 1a and b*). Collectively, robust and specific co-expression of GDNF receptors in cluster 1, together with the prediction of RTK activation and involvement of GDNF signaling pathway downstream TFs in clusters 2 and 3, suggests that GDNF signaling could be the mechanism by which FAPs sense the distant nerve injury, where local Schwann cells act as the GDNF source upon nerve injury.

## The GDNF-BDNF axis as FAP's response mechanism to nerve injury

Next, to discover how FAPs may contribute to nerve regeneration, we screened the list of genes enriched in clusters 2 and 3 that were predicted to be downstream of the GDNF signaling pathway to identify candidate effector genes. From the TRRUST analysis results, 44 genes were identified to be regulated by either *Fos*, *Jun*, or NF-κB (*Figure 5A*, *Supplementary file 1a, b and c*). Since FAPs themselves do not constitute the neural components within skeletal muscle, we reasoned that secreted factors from FAPs would most likely exert a beneficial effect on the regenerating nerves. Also, considering the effector gene's potential function in supporting nerve regeneration, we presumed that it regulates neurons or glial cells. Moreover, we anticipated that expression of the effector gene would be limited to the context of nerve injury and regeneration, since the vast majority of FAPs are in a quiescent state in unperturbed adult muscles (*Scott et al., 2019*). Thus, we applied the following criteria to narrow down our candidate gene list: (1) genes that are known to code secreted proteins, (2) genes that are known to regulate neurons or glial cells, and (3) genes that are expressed exclusively in activated FAPs in response to nerve injury (*Figure 5A*, *Supplementary file 1c*). Unexpectedly, after filtering out genes that did not fit the three criteria, only *Bdnf* remained in our candidate gene list that could act as the effector secreted by FAPs upon nerve injury to support nerve regeneration (*Figure 5A*). Expression patterns of *Bdnf* in FAPs upon nerve injury showed transient upregulation in SNC-affected FAPs, whereas chronic expression of *Bdnf* was observed in DEN-affected FAPs, showing correlation with its potential requirement during nerve regeneration (*Figure 5B*). The expression of *Bdnf* was mostly limited to cluster 2 (*Figure 5B and C*), where pathway analysis and TF prediction suggested the involvement of GDNF-RET-Ras-ERK-Fos signaling cascade in this subset of FAPs (*Figure 4D–F*, *Figure 4—figure supplements 1–4*). Accordingly, we hypothesized that FAPs secrete BDNF in response to GDNF from Schwann cells upon nerve injury, to actively take part in the regeneration process.

To validate our hypothesis in vivo, we first examined the expression profiles of *Gdnf* in Schwann cells and *Bdnf* in FAPs at early time points in response to SNC, using *Plp1^{CreER}; Rosa26^{LSL-tdTomato}* mice to specifically label and hence isolate Schwann cells (*Doerflinger et al., 2003*; *Figure 5D*). Expectedly, we could observe sequential upregulation of *Gdnf* and *Bdnf* from Schwann cells and FAPs, respectively, where *Gdnf* levels peaked at 1 dpi in Schwann cells, followed by a gradual increase of *Bdnf* expression in FAPs, which peaked at 3 dpi (*Figure 5E*). Following mRNA expression validation in vivo, we performed western blot analysis using FAPs isolated from either SNC-affected muscles at 7 dpi or from the contralateral, uninjured muscles to validate the expression of BDNF protein upon nerve injury (*Figure 5—figure supplement 1*). FAPs from both uninjured muscles and SNC-affected muscles showed robust expression of PDGFRα, a well-known marker for FAPs (*Joe et al., 2010*; *Uezumi et al., 2010*), indicating successful isolation and protein extraction from the sorted FAPs (*Figure 5F*). However, unlike PDGFRα, the mature form of BDNF protein could only be detected in SNC-affected FAPs, but not in uninjured FAPs, showing correlative results with the mRNA expression pattern of *Bdnf* in FAPs upon nerve injury (*Figure 5E and F*). These results demonstrate the expression of BDNF in nerve injury-affected FAPs, but not in uninjured FAPs, on both mRNA and protein levels.

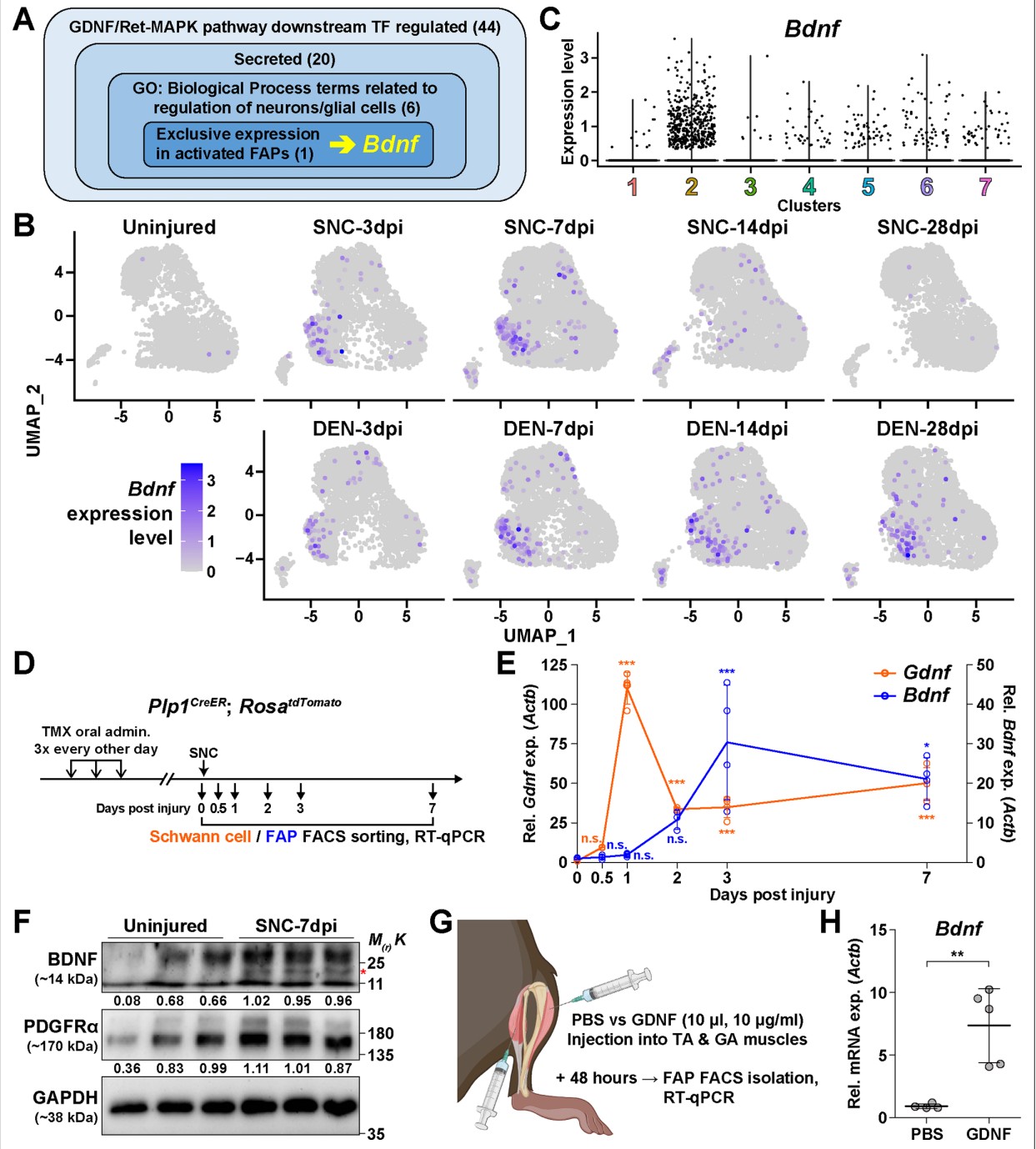

**Figure 5.** The GDNF-BDNF axis as fibro-adipogenic progenitors (FAPs) response mechanism to nerve injury. (**A**) Identification of candidate genes expressed in FAPs in response to glial cell line-derived neurotrophic factor (GDNF) that may contribute to nerve regeneration. Number of genes that fit into each criterion is indicated. (**B**) Expression of *Bdnf* in each scRNA-seq sample shown on uniform manifold approximation and projection (UMAP) plots. (**C**) Violin plot displaying the expression levels of *Bdnf* in the seven FAP clusters. (**D**) Scheme for sampling Schwann cells and FAPs at different time points post sciatic nerve crush (SNC) for gene expression analyses. (**E**) RT-qPCR results showing expression levels of *Gdnf* in Schwann cells (orange dot and line, left y-axis) and *Bdnf* in FAPs (blue dot and line, right y-axis) at different time points post-SNC. n=4, except for 0 and 2 dpi, where n=3. One-way ANOVA with Bonferroni's post hoc test. *p<0.05, ***p<0.001, n.s., not significant. (**F**) Western blot results showing BDNF protein expression in PDGFRα+ FAPs isolated from SNC-affected or uninjured contralateral muscles at 7 dpi. n=3. Mature form of BDNF is indicated with a red asterisk. Quantified values normalized to GAPDH is indicated below each protein. See *Figure 5—figure supplement 1* for the experimental scheme. (**G**) Scheme for intramuscular injection of either PBS or recombinant mouse GDNF protein, with the time point for FAP isolation post-injection indicated. (**H**) RT-qPCR

*Figure 5 continued on next page*

*Figure 5 continued*

results show the expression level of *Bdnf* in FAPs 48 hr post intramuscular injection of either PBS (n=4) or GDNF (n=5). Unpaired t-test with Welch's correction. **p<0.01.

The online version of this article includes the following source data and figure supplement(s) for figure 5:

**Source data 1.** The zip file contains raw western blot images, a marker image, and a marker-merged, labeled image obtained for *Figure 5F*.

**Figure supplement 1.** Scheme for western blot analysis.

**Figure supplement 2.** Decreased intramuscular glial cell line-derived neurotrophic factor (GDNF) activity can weaken *Bdnf* induction in fibro-adipogenic progenitors (FAPs) upon nerve injury.

Next, to validate the sufficiency of GDNF signaling in inducing *Bdnf* expression in FAPs in vivo, we injected recombinant mouse GDNF protein into the TA and the two gastrocnemius (GA) muscles (lateral and medial GA), from which FAPs were FACS-isolated 48 hr post-injection to investigate the expression of *Bdnf* (*Figure 5G*). Compared to PBS control, intramuscular injection of GDNF sufficiently induced *Bdnf* expression in FAPs, even in the absence of a nerve injury (*Figure 5H*). Conversely, to show the necessity of GDNF signaling in the upregulation of *Bdnf* in FAPs upon nerve injury, we injected either IgG control or anti-GDNF antibodies into the TA and GA muscles 24 hr after SNC, and FACS-isolated FAPs 48 hr post-injection for analysis. Though the results were not statistically significant, injection of GDNF-blocking antibodies showed a tendency to reduce *Bdnf* expression compared to IgG-injected controls, which is in support of our hypothesis (*Figure 5—figure supplement 2*). Together, we suggest that FAPs can respond to nerve injury via the GDNF-BDNF axis, since recombinant GDNF protein could sufficiently induce *Bdnf* expression in FAPs without nerve injury, and reduced GDNF activity could weaken *Bdnf* expression in the nerve injury-affected FAPs.

## Remyelination by FAP-derived BDNF during peripheral nerve regeneration

Although BDNF is known to function in processes such as axon elongation (*Oudega and Hagg, 1999*; *English et al., 2013*), survival of neurons (*Ghosh et al., 1994*; *Baydyuk and Xu, 2014*), and myelination by Schwann cells (*Zhang et al., 2000*; *Chan et al., 2001*; *Xiao et al., 2009*), the role of BDNF secreted by FAPs in nerve regeneration is unknown. To find out how FAP-derived BDNF can contribute to nerve regeneration, we produced conditional knockout (cKO) mice where *Bdnf* is specifically inactivated in mesenchymal progenitors including FAPs, by crossing *Prrx1*^*Cre* mice (*Logan et al., 2002*; *Kim et al., 2022*; *Leinroth et al., 2022*; *Hann et al., 2024*) with *Bdnf*-floxed (*Bdnf*^*fl*) mice (*Prrx1*^*Cre*; *Bdnf*^*fl/fl*, hereafter cKO) (*Figure 6A*, *Figure 6—figure supplement 1A*). Inactivation of *Bdnf* in FAPs in the cKO mice was confirmed on both genomic DNA and mRNA levels (*Figure 6—figure supplement 1B, C*). Though Cre expression in *Prrx1*-expressing cells occurs from embryonic day 9.5 (*Logan et al., 2002*), no visible phenotypes were observed in the postnatal, juvenile, and adult cKO mice compared to littermate controls (hereafter, Ctrl). However, upon SNC in the right hindlimb, cKO mice displayed a delay in nerve regeneration compared to Ctrl, measured by compound muscle action potential (CMAP) amplitude and latency via electromyography (EMG) on the GA muscle (*Figure 6A–D*, *Figure 6—figure supplement 1D, E*). At 4 weeks post-injury (wpi), nerve regeneration in both Ctrl and cKO mice showed insufficient recovery in the right, injured GA compared to the left, uninjured GA, where lower amplitude and prolonged latency in CMAP was observed (*Figure 6B–D*). However, at 6 wpi, while CMAP amplitude and latency in the left and right GAs became comparable in Ctrl mice, recovery of such values were stalled at levels comparable to 4 wpi in the cKO mice (*Figure 6B–D*). By 12 wpi, electrophysiological functions of the injured nerves became statistically comparable to that of its contralateral counterpart in the cKO mice, indicating a delayed regeneration in the cKO mice (*Figure 6B–D*).

Generally, a decrease in CMAP amplitude and prolonged CMAP latency can be explained by two main causes: axonal loss and defective myelination (*Mallik and Weir, 2005*; *Chung et al., 2014*). Since complete regeneration of the injured nerves on the electrophysiological level could be achieved after a sufficient period of time in the cKO mice (*Figure 6B–D*), we presumed that axonal loss would not have occurred, since it would result in permanent defects by loss of motor units. Instead, we thought

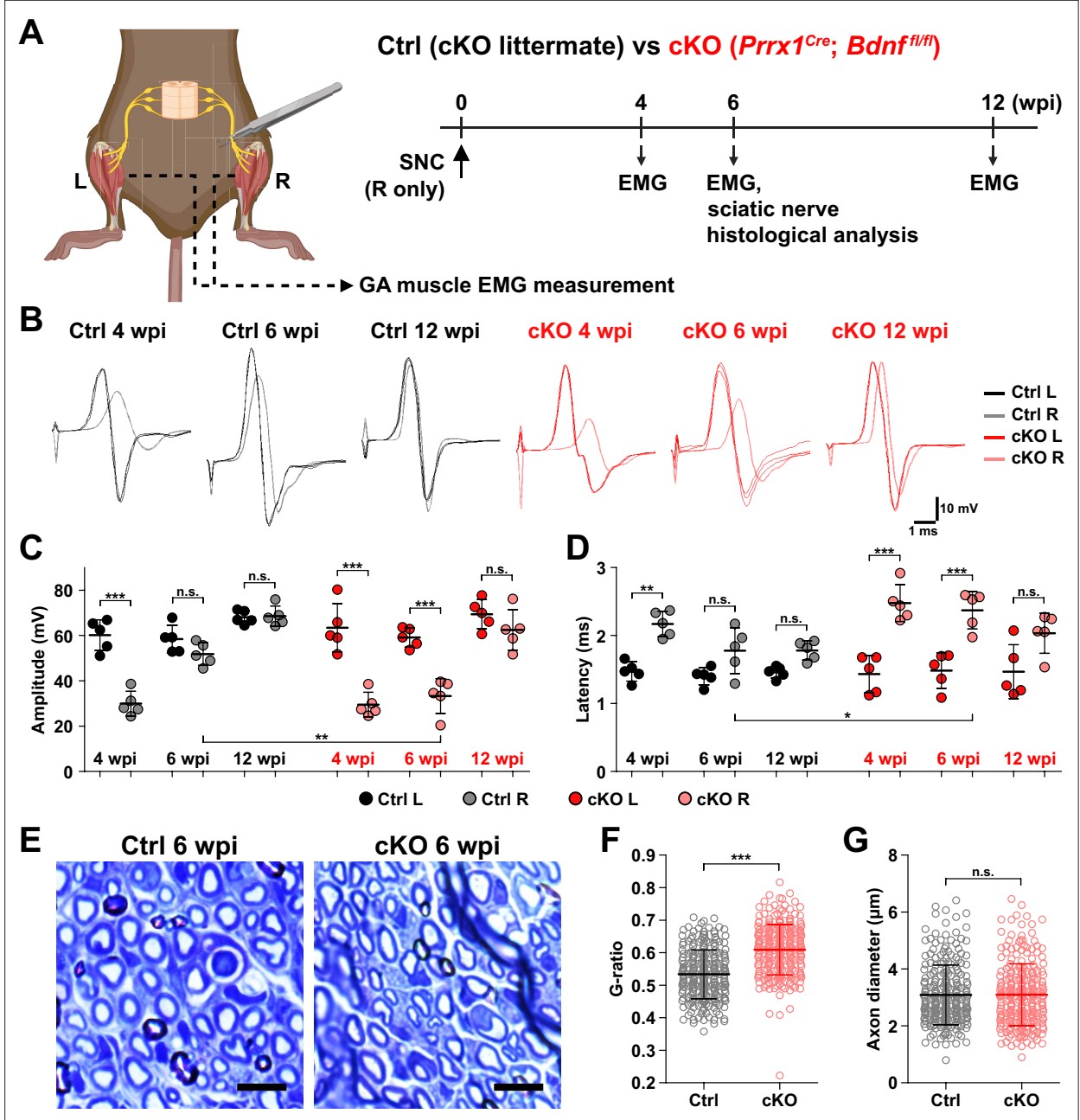

**Figure 6.** Remyelination by fibro-adipogenic progenitor (FAP)-derived BDNF during peripheral nerve regeneration. (**A**) Experimental scheme displaying mice used and the time points selected for electromyography (EMG) measurements and sciatic nerve dissection. wpi, weeks post-injury. (**B**) Representative EMG measurement results of both injured and uninjured gastrocnemius (GA) muscles from Ctrl or conditional knockout (cKO) mice at the indicated time points post-sciatic nerve crush (SNC). (**C, D**) Quantified results of EMG measurement showing (**C**) compound muscle action potential (CMAP) amplitude and (**D**) CMAP latency. n=5. One-way ANOVA with Bonferroni's post hoc test. *p<0.05, **p<0.01, ***p<0.001, n.s., not significant. (**E**) Representative images showing toluidine blue-stained, semi-thin cross-sections of sciatic nerves dissected from Ctrl or cKO mice at 6 wpi. Scale bars, 10 μm. (**F–G**) Quantification of (**F**) calculated G-ratio values and (**G**) axon diameters from analyzing toluidine blue-stained sciatic nerve sections dissected from Ctrl or cKO mice at 6 wpi. 50 axons were randomly selected from each sciatic nerve for quantification. n=5. Mann-Whitney U test. ***p<0.001, n.s., not significant.

The online version of this article includes the following source data and figure supplement(s) for figure 6:

**Figure supplement 1.** Validation of conditional knockout (cKO) mice used in this study and methods used for analysis.

*Figure 6 continued on next page*

*Figure 6 continued*

**Figure supplement 1—source data 1.** The zip file contains raw DNA electrophoresis gel images and labeled images obtained for *Figure 6—figure supplement 1B*.

that defective myelination could have occurred in the cKO mice, considering the fact that BDNF is already known to promote remyelination during peripheral nerve regeneration (*Zhang et al., 2000*; *Chan et al., 2001*; *Zheng et al., 2016*), and that defective myelination alone can affect both CMAP amplitude and latency (*Mallik and Weir, 2005*). Thus, we investigated the effect of conditional *Bdnf* inactivation in FAPs on regenerative myelination by examining the sciatic nerves from Ctrl versus cKO mice at 6 wpi, when the delayed functional recovery of the injured nerves in the cKO mice was prominent (*Figure 6A–D*). Toluidine blue staining of the semi-thin sections of injured sciatic nerves revealed significantly reduced myelin thickness in the cKO mice compared to Ctrl mice (*Figure 6E*). Indeed, higher G-ratio values were calculated from cKO mice compared to Ctrl, confirming the reduced myelination in the regenerating nerves in cKO mice (*Figure 6F*, *Figure 6—figure supplement 1F–H*). This decrease in myelin thickness was independent from axon diameter, which were comparable in both Ctrl and cKO mice, implying that no axonal loss or defect had occurred in the cKO mice compared to controls (*Figure 6G*). Taken together, our results revealed the direct involvement of FAP-derived BDNF in the remyelination process during peripheral nerve regeneration, such that inadequate levels

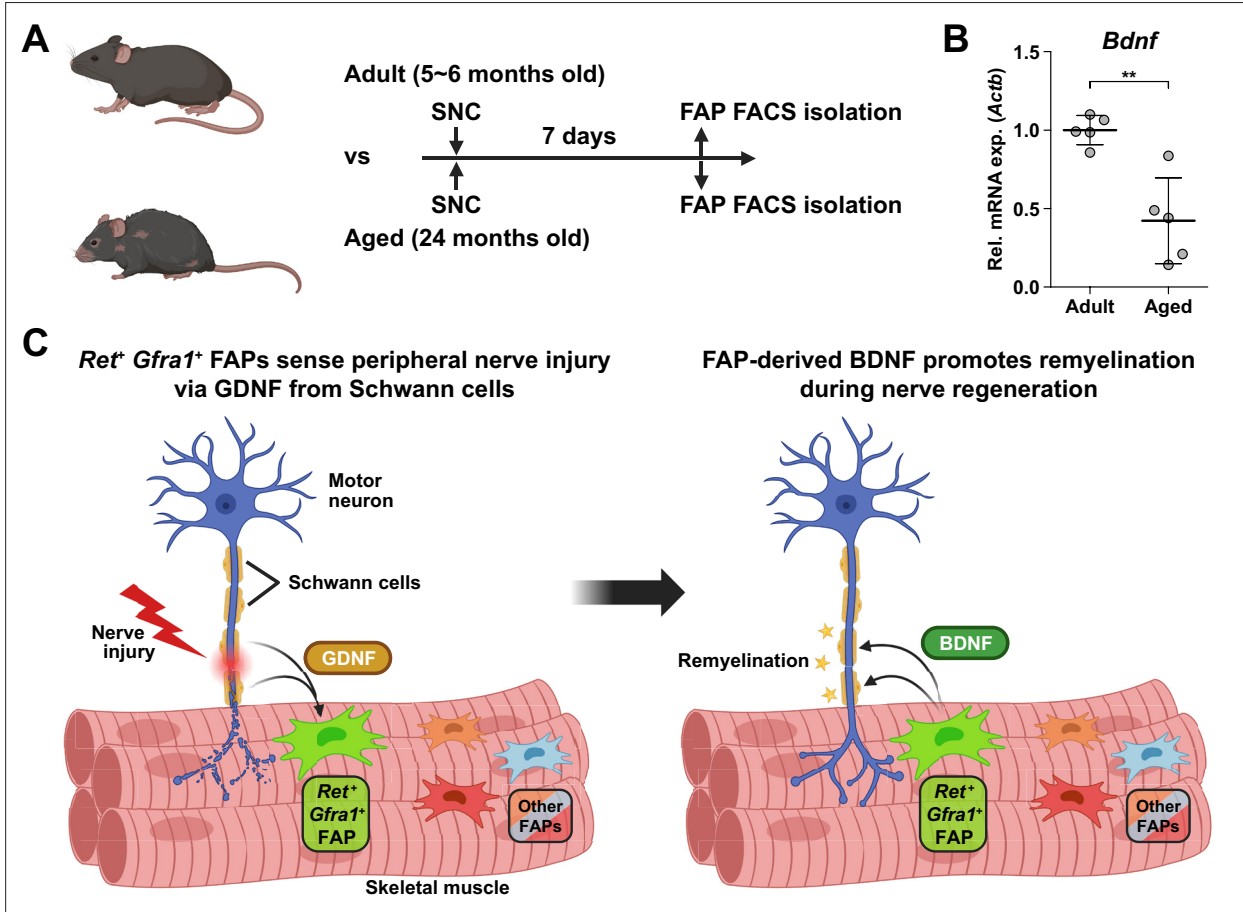

**Figure 7.** Implication of fibro-adipogenic progenitor (FAP)-derived BDNF in the age-related delay in nerve regeneration. (**A**) Experimental scheme indicating the ages of mice used and the time point for FAP isolation to compare the expression level of *Bdnf* post-sciatic nerve crush (SNC). (**B**) RT-qPCR results show the expression level of *Bdnf* in FAPs isolated from either adult (5–6 months) or aged (24 months) mice at 7 dpi. n=5. Unpaired t-test. **p<0.01. (**C**) Graphical summary of this study.

of *Bdnf* expression in FAPs caused delayed remyelination and hence delayed nerve regeneration in the cKO mice.

## Implication of FAP-derived BDNF in the age-related delay in nerve regeneration

Finally, to seek the clinical relevance of *Bdnf* expression in FAPs during nerve regeneration, we compared the expression levels of *Bdnf* in adult (5–6 months) versus aged (24 months) mice in FAPs post-SNC. At 7 dpi, *Bdnf* expression was significantly reduced in the aged mice compared to adult mice (*Figure 7A and B*). Such a difference could be one of the factors that lead to the delayed or failed regeneration of injured nerves in the elderly compared to healthy, young adults, suggesting the clinical role of FAPs in a timely, successful peripheral nerve regeneration process.

## Discussion

Traumatic injury to the peripheral nerve has severe consequences, including lifelong paralysis of the injured limb that can compromise the quality of life significantly (*Grinsell and Keating, 2014*). Thus, understanding the regeneration process of the peripheral nerves is fundamental for treating the potentially devastating injury. Previously, several cellular components in and outside the injured nerve were discovered to actively participate in the regeneration process, including the injured neurons (*Hanz et al., 2003*), glial cells (*Arthur-Farraj et al., 2012*), immune cells (*Mueller et al., 2001*; *Kalinski et al., 2020*), and nerve-resident mesenchymal cells (*Parrinello et al., 2010*; *Toma et al., 2020*), via diverse mechanisms so that the nerve can regain its function (*Scheib and Höke, 2013*). In this study, we investigated the response mechanism of muscle-resident FAPs to both acute and chronic peripheral nerve injury via scRNA-seq, and revealed that this population of cells can also actively take part in the nerve regeneration process. Here, we discovered that muscle-resident FAPs can recognize the distant nerve injury by sensing GDNF secreted by Schwann cells. Though GDNF secretion by Schwann cells in response to nerve injury had previously been recognized (*Hammarberg et al., 1996*; *Höke et al., 2002*; *Arthur-Farraj et al., 2012*; *Xu et al., 2013*; *Proietti et al., 2021*), we identified FAPs as a major target cell population of GDNF within skeletal muscle, based on the enriched expression of GDNF receptors. In-depth, exploiting the technical advantage of scRNA-seq, we suggested that a subset of FAPs, named cluster 1 in this study, can sense the local GDNF by expressing GDNF receptors *Ret* and *Gfra1*, and that upon GDNF sensing, cluster 1 FAPs turn into clusters 2 and 3 to contribute to the nerve regeneration process. Specifically, we discovered that FAPs, especially the *F2rl1*-expressing cluster 2, express *Bdnf* in response to nerve injury and/or GDNF, which in turn was shown to promote the remyelination process by Schwann cells during nerve regeneration, using our cKO mouse model (*Figure 7C*). Since epineurial and perineurial, but not endoneurial mesenchymal cells share their origins with the muscle-resident FAPs and, therefore, are *Prrx1*-positive (*Joseph et al., 2004*; *Carr et al., 2019*), the possibility that the delayed remyelination observed in our cKO mice could be due to the combined effect of *Bdnf* depletion in both muscle- and nerve-resident mesenchymal cells cannot be eliminated. To resolve such an issue, further investigations using muscle-resident FAP-specific Cre mouse lines are required, but such a line is currently unavailable as no such specific marker has been found. Nevertheless, our findings show that muscle-resident FAP-derived BDNF is indeed important for nerve regeneration, since intramuscular injection of recombinant BDNF can sufficiently accelerate the nerve regeneration process (*Zheng et al., 2016*). Conversely, intramuscular injection of BDNF-neutralizing antibodies can sufficiently delay nerve regeneration (*Zheng et al., 2016*). Thus, endogenous supply of intramuscular BDNF by the muscle-resident FAPs in our Ctrl mice would likely have supported remyelination by Schwann cells, while the lack of such BDNF supply by FAPs in our cKO mice would have resulted in delayed remyelination. Also, we found that while muscle-resident FAPs robustly express both GDNF receptor genes, neither epineurial nor perineurial mesenchymal cells express significant levels of *Ret* and *Gfra1* (*Carr et al., 2019*; *Toma et al., 2020*; *Zhao et al., 2022*; *Figure 8*), implying that the GDNF-BDNF axis found in this study could be valid uniquely in muscle-resident FAPs. Collectively, we suggest that muscle-resident mesenchymal progenitors can directly contribute to nerve regeneration via the GDNF-BDNF axis (*Figure 7C*), of which was previously unidentified; we suggest that in future studies regarding peripheral nerve regeneration, active participation of this intramuscular mesenchymal population should be taken into consideration.

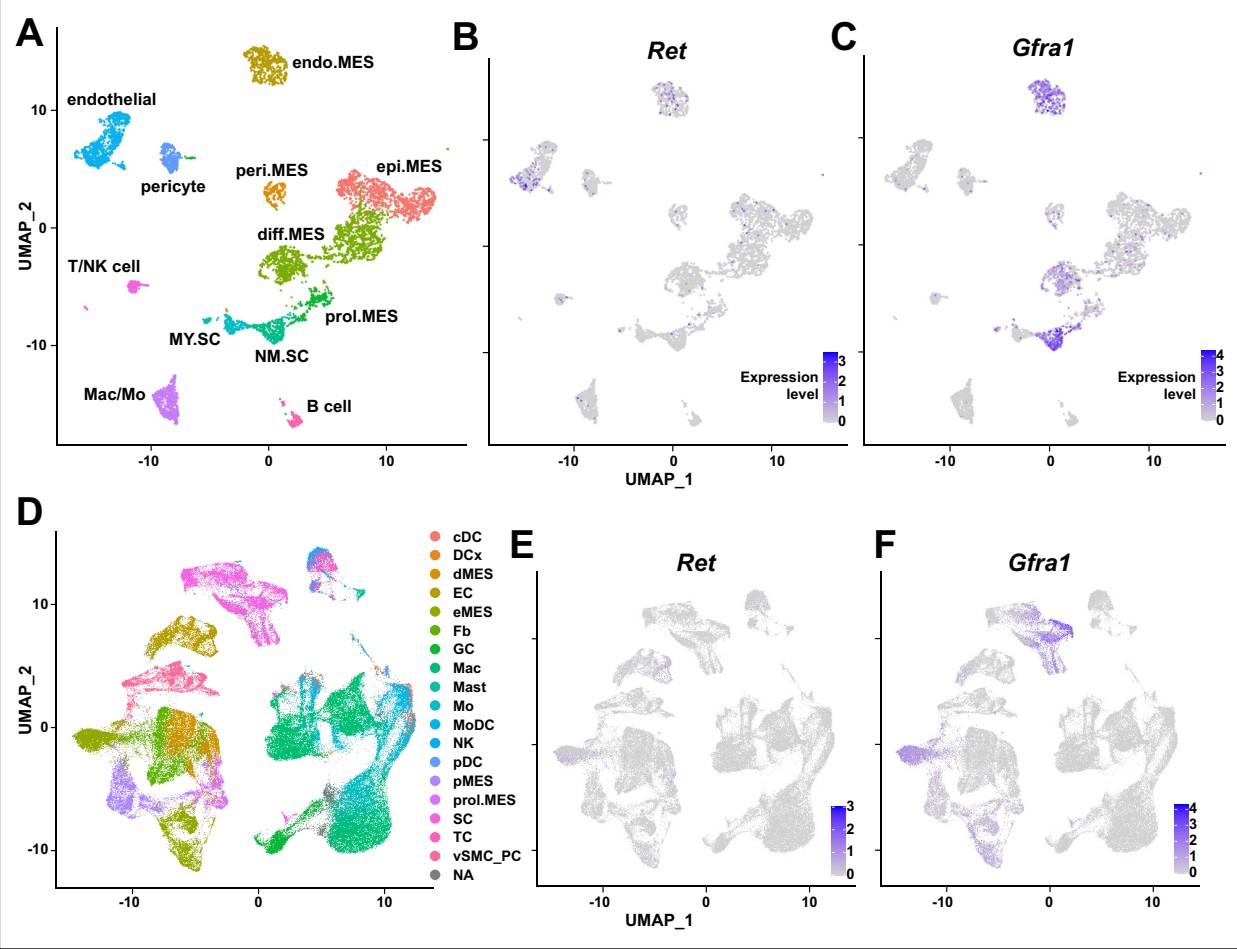

**Figure 8.** Expression of glial cell line-derived neurotrophic factor (GDNF) receptor genes *Ret* and *Gfra1* in nerve-resident cells. (**A–C**) Single-cell RNA-sequencing (scRNA-seq) data from *Carr et al., 2019* and *Toma et al., 2020* (accession numbers: GSM3408137, GSM3408139, GSM4423509, and GSM4423506) were merged into a single Seurat object and visualized on uniform manifold approximation and projection (UM)AP plots. Cell types identified using markers listed by *Toma et al., 2020* are shown in (**A**), and expression levels of (**B**) *Ret* and (**C**) *Gfra1* are displayed. epi.MES: epineurial mesenchymal cells; peri.MES: perineurial mesenchymal cells; endo.MES: endoneurial mesenchymal cells; diff.MES: differentiating mesenchymal cells; prol.MES: proliferating mesenchymal cells; NM.SC: non-myelinating Schwann cells; MY.SC: myelinating Schwann cells; Mac/Mo: macrophage/monocyte. (**D–F**) scRNA-seq data from *Zhao et al., 2022* (accession number: GSE198582) were merged into a single Seurat object and visualized on UMAP plots. Cell types annotated by *Zhao et al., 2022* are shown in (**D**), and expression levels of (**E**) *Ret* and (**F**) *Gfra1* are displayed. cDC: conventional dendritic cells; DCx: dendritic cells destined for homing; dMES: differentiating mesenchymal cells; EC: endothelial cells; eMES: endoneurial mesenchymal cells; Fb: fibroblasts; GC: granulocytes; Mac: macrophages; Mast: mast cells; Mo: monocytes; MoDC: monocyte-derived dendritic cells; NK: natural killer cells; pDC: plasmocytoid dendritic cells; pMES: perineurial mesenchymal cells; prol.MES: proliferating mesenchymal cells; SC: Schwann cells; TC: T cells; vSMC_PC: vascular smooth muscle cells/pericytes; NA: not applicable.

In that process, our scRNA-seq data may provide valuable insights, which may lead to additional discoveries on FAP's contributions to nerve regeneration by other mechanisms, and/or provide clues on the cell-to-cell communications of FAPs with other cell types that may lead to facilitation of the regeneration process.

Here, we have primarily investigated the roles of *Ret*-expressing and *F2rl1*-expressing FAPs in sensing and responding to nerve injury, respectively. However, possibilities that other subpopulations can exert distinct beneficial effects on nerve regeneration remain largely unexplored. For example, although identified as a nerve injury-relevant subpopulation in this study, specific contributions of the *Alkal2*-expressing cluster 3 FAPs to nerve regeneration are yet to be discovered. Identification of effector genes such as *Bdnf* from this subpopulation may lead to the discovery of an additional mechanism by which FAPs can contribute to nerve regeneration. Although other subpopulations (clusters 4–7) did not exhibit dramatic fluctuations in population percentage, some of the genes expressed

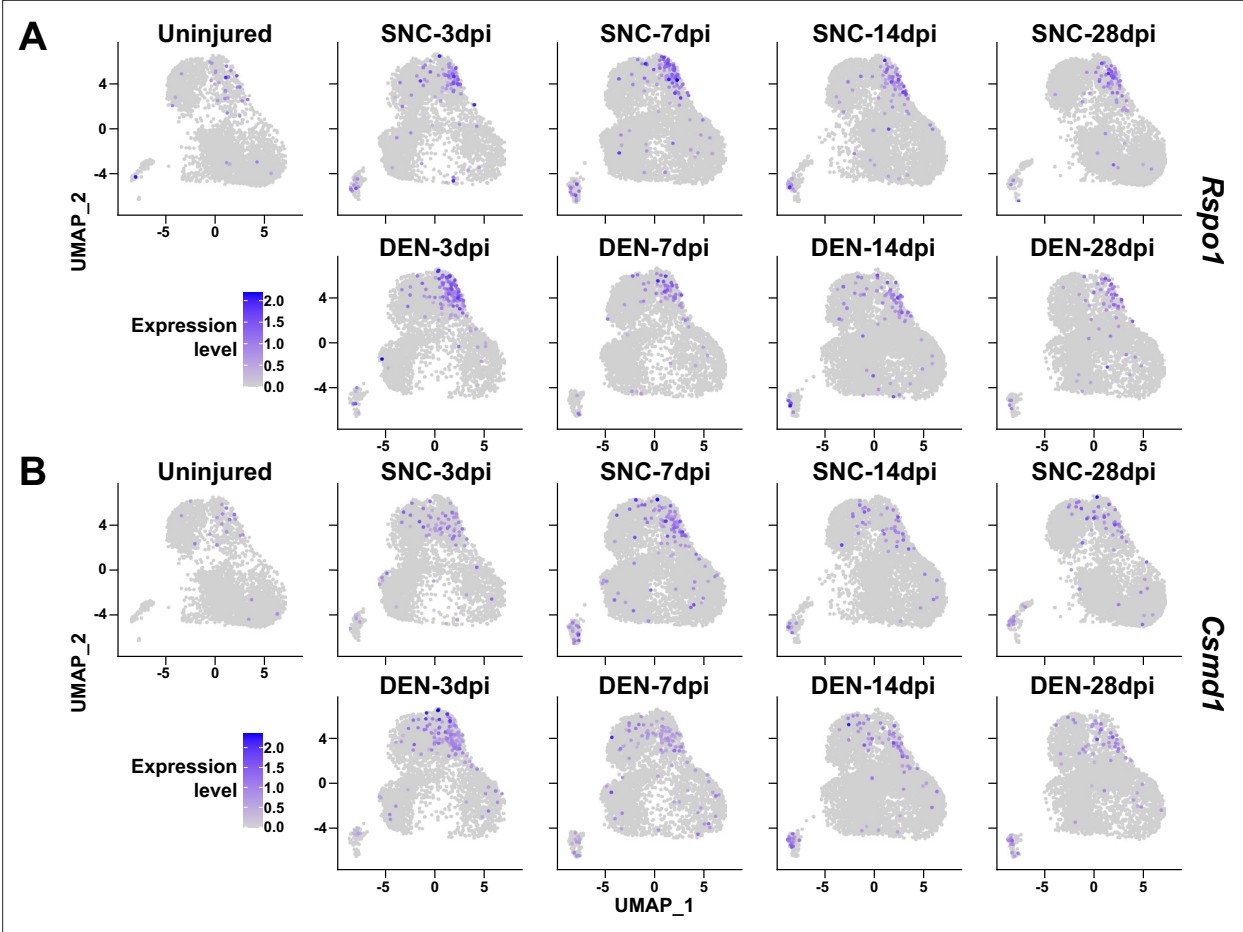

**Figure 9.** Expression of nerve injury-induced, cluster-specific genes in fibro-adipogenic progenitors (FAPs). (**A, B**) Expressions of (**A**) *Rspo1* and (**B**) *Csmd1* are shown on uniform manifold approximation and projection (UMAP) plots, separated by samples. Single-cell RNA-sequencing (scRNA-seq) data obtained in this study were used.

specifically in those subpopulations showed patterns that followed the regeneration process (*Figure 9*). Such gene expression patterns suggest subpopulation-specific functions even in the less dynamic subpopulations within FAPs in peripheral nerve regeneration, and require further investigations to reveal their possible contributions.

The role of BDNF in peripheral nerve regeneration has been identified previously, where it was found to promote both intrinsic axonal regeneration in neurons as well as remyelination by Schwann cells (*Zhang et al., 2000*; *Chan et al., 2001*; *Zheng et al., 2016*). In such circumstances, various cellular sources of BDNF have been identified. Bone marrow transplantation of wild-type cells into *Bdnf* heterozygotic knockout mice revealed the involvement of bone marrow-derived cells in expressing *Bdnf* that can promote nerve regeneration in the sciatic nerve (*Takemura et al., 2012*). Schwann cells themselves are cellular sources of BDNF during nerve regeneration (*Wilhelm et al., 2012*). In this study, we showed that BDNF from FAPs can also promote myelination of the regenerating axons post-injury, suggesting FAPs as an additional cellular source of BDNF in peripheral nerve regeneration. The existence of cellular sources of BDNF other than FAPs such as Schwann cells would provide an explanation for the delayed, but not failed, remyelination in our *Bdnf* cKO mice, where complete regeneration had occurred after a sufficient amount of time, despite the lack of *Bdnf* expression in FAPs. Still, ablation of *Bdnf* in FAPs displayed significant delays in the remyelination process during nerve regeneration, suggesting their requirement in the timely regeneration process of injured nerves. Meanwhile, scRNA-seq data of all mononuclear cells from denervated muscles (*Nicoletti et al., 2023*) suggested expression of *Bdnf* in tenocytes and pericytes in addition to Schwann cells and FAPs (*Figure 10A and B*). Although this may imply the involvement of those cellular components in providing BDNF

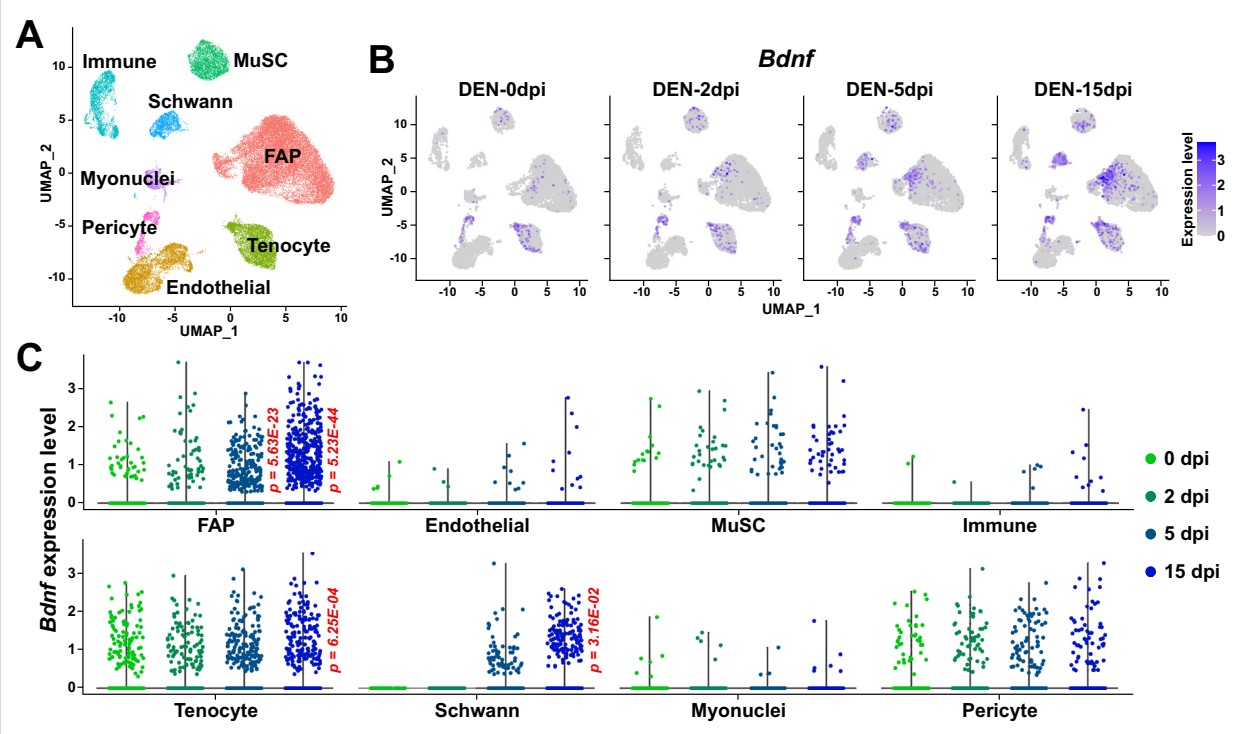

**Figure 10.** Expression of *Bdnf* in muscle-resident mononuclear cells affected by denervation. (**A**) Uniform manifold approximation and projection (UMAP) plot showing data from *Nicoletti et al., 2023* labeled by cell types identified. (**B**) Expression pattern of *Bdnf* in data from *Nicoletti et al., 2023* shown on UMAP plots, separately by days post denervation. (**C**) Expression of *Bdnf* in each cell type on different days post-denervation displayed in violin plots. p-values were calculated by comparing each injury-affected cells' expression levels versus its uninjured state (0 dpi). Only significant p-values are shown. Wilcoxon rank sum test.

for nerve regeneration, expression of *Bdnf* in such cells were not altered as significantly in response to nerve injury as in FAPs or Schwann cells (*Figure 10C*). Moreover, while *Ret*-expressing FAPs and Schwann cells are known to be in proximity to the NMJ or relevant nerve regeneration sites (*Leinroth et al., 2022*), such locational enrichment of tenocytes and pericytes have not been reported so far. Thus, it is likely that FAPs, together with Schwann cells, are the main sources of BDNF within skeletal muscle that can act in the remyelination process during peripheral nerve regeneration.

Aging is one of the well-known factors that can slow down the nerve regeneration process (*Verdú et al., 2000*; *Maita et al., 2023*). Since multiple cell types are known to participate in this process (*Scheib and Höke, 2013*), the determination of the cell types that can cause age-related delays in nerve regeneration is important for the development of therapeutic approaches targeting the relevant cell types. Of note, previous research emphasized the importance of niche factors, rather than the intrinsic regenerative capacity of the injured neurons, in the age-related decline in nerve regeneration (*Painter et al., 2014*). Specifically, the inability of Schwann cells to adopt repair cell phenotypes has been pointed out as one of the age-related changes (*Painter et al., 2014*; *Wagstaff et al., 2021*). In addition, age-related changes in immune cells, especially macrophages, were suggested as causal factors that delay nerve regeneration (*Büttner et al., 2018*). In particular, chronic inflammatory phenotypes were shown to interfere with the remyelination process by Schwann cells (*Büttner et al., 2018*), and failed macrophage infiltration in the early stages of regeneration resulted in defective Wallerian degeneration and myelin debris clearing (*Scheib and Höke, 2016*). In addition to the involvement of Schwann cells and immune cells, we have identified muscle-resident FAPs as an additional cellular component that can contribute to nerve regeneration, by promoting remyelination via BDNF secretion. Surprisingly, expression of *Bdnf* by FAPs was significantly reduced in aged mice compared to adult mice, suggesting the clinical relevance of FAP's involvement in the age-related delay in peripheral nerve regeneration. We believe that further studies on age-related changes in FAPs may provide valuable clues to understanding clinical observations from aged individuals, which can lead to the

development of additional therapeutic strategies that include FAPs as target cells in treating both young and aged nerve injury patients.

# Materials and methods

## Key resources table

| Reagent type (species) or resource | Designation | Source or reference | Identifiers | Additional information |
|---|---|---|---|---|
| Antibody | APC anti-mouse CD31 (Rat monoclonal) | BioLegend | Cat# 102510, RRID:AB_312917 | FACS, 1:100 |
| Antibody | APC anti-mouse CD45 (Rat monoclonal) | BioLegend | Cat# 103112, RRID:AB_312977 | FACS, 1:100 |
| Antibody | PE anti-mouse TER-119 (Rat monoclonal) | Biolegend | Cat# 116208, RRID:AB_313709 | FACS, 1:100 |
| Antibody | Biotin anti-mouse CD106 (Vcam1) (Rat monoclonal) | Biolegend | Cat# 105704, RRID:AB_313205 | FACS, 1:100 |
| Antibody | FITC anti-mouse Ly-6A/E (Sca1) (Rat monoclonal) | BD Pharmingen | Cat# 553335, RRID:AB_394791 | FACS, 1:100 |
| Antibody | anti-GAPDH (rabbit polyclonal) | Bethyl Laboratories | Cat# A300-641A, RRID:AB_513619 | WB, 1:1000 |
| Antibody | anti-PDGFRα (rabbit polyclonal) | Santa Cruz Biotechnology | Cat# sc-338, RRID:AB_631064 | WB, 1:200 |
| Antibody | anti-BDNF (rabbit polyclonal) | Alomone Labs | Cat# ANT-010, RRID:AB_2039756 | WB, 1:1000 |
| Antibody | anti-GDNF (rabbit polyclonal) | Alomone Labs | Cat# ANT-014, RRID:AB_2039876 | 10 µg per injection |
| Antibody | Normal rabbit IgG (polyclonal) | Sino Biological | Cat# CR1, RRID:AB_3073921 | 10 µg per injection |
| Antibody | horseradish peroxidase-conjugated anti-rabbit IgG (goat polyclonal) | Promega | Cat# W4011, RRID:AB_430833 | 1:10,000 |
| Chemical compound, drug | Tamoxifen | Sigma-Aldrich | Cat# T5648 | |
| Chemical compound, drug | 2,2,2-Tribromoethanol (Avertin) | Sigma-Aldrich | Cat# T48402 | |
| Chemical compound, drug | 7-aminoactinomycin D (7-AAD) | Sigma-Aldrich | Cat# SML1633 | 1:1000 |
| Chemical compound, drug | PE/Cyanine7 Streptavidin | Biolegend | Cat# 405206 | 1:100 |
| Chemical compound, drug | TRIzol Reagent | Invitrogen | Cat# 15596–018 | |
| Chemical compound, drug | Paraformaldehyde | Sigma-Aldrich | Cat# P6148 | |
| Chemical compound, drug | Sodium cacodylate buffer | Electron Microscopy Sciences | Cat# 11652 | |
| Chemical compound, drug | Osmium tetroxide | Electron Microscopy Sciences | Cat# 19190 | |
| Chemical compound, drug | Uranyl acetate solution | Electron Microscopy Sciences | Cat# 22400 | |
| Chemical compound, drug | Propylene oxide | Tokyo Chemical Industry | Cat# E0016 | |
| Chemical compound, drug | Spurr's resin | Electron Microscopy Sciences | Cat# 14300 | |
| Chemical compound, drug | Toluidine blue | Sigma-Aldrich | Cat# 89640 | |
| Chemical compound, drug | Sodium borate | Sigma-Aldrich | Cat# B9876 | |
| Strain, strain background (*Mus musculus*) | wild type B6: C57BL/6 J | The Jackson Laboratory | RRID:IMSR_JAX:000664 | |

*Continued on next page*

*Continued*

| Reagent type (species) or resource | Designation | Source or reference | Identifiers | Additional information |
|---|---|---|---|---|
| Strain, strain background (*Mus musculus*) | *Prrx1$^{Cre}$*: B6.Cg-Tg(Prrx1-cre)1Cjt/J | The Jackson Laboratory | RRID:IMSR_JAX:005584 | |
| Strain, strain background (*Mus musculus*) | *Bdnf$^{fl}$*: Bdnf$^{tm3Jae}$/J | The Jackson Laboratory | RRID:IMSR_JAX:004339 | |
| Strain, strain background (*Mus musculus*) | *Plp1$^{CreER}$*: B6.Cg-Tg(Plp1-cre/ERT)3Pop/J | The Jackson Laboratory | RRID:IMSR_JAX:005975 | |
| Strain, strain background (*Mus musculus*) | *Rosa26$^{LSL-tdTomato}$*: B6.Cg-Gt(ROSA)26Sor$^{tm14(CAG-tdTomato)Hze}$/J | The Jackson Laboratory | RRID:IMSR_JAX:007914 | |
| Peptide, recombinant protein | Recombinant mouse GDNF | Sigma-Aldrich | Cat# SRP3200 | 10 µg/ml, 10 µl per injection |
| Commercial assay or kit | Chromium Next GEM Single Cell 3' Kit v3.1 | 10 X Genomics | Cat# PN-1000268 | |
| Commercial assay or kit | ReverTra Ace qPCR RT Master Mix | Toyobo | Cat# FSQ-201 | |
| Commercial assay or kit | ORA SEE qPCR Green ROX L Mix | HighQu | Cat# QPD0550 | |
| Software, algorithm | CellRanger v3.1.0 | 10 X Genomics | RRID:SCR_023221 | |
| Software, algorithm | Velocyto v0.17 | *La Manno et al., 2018* | RRID:SCR_018167 | |
| Software, algorithm | R package Seurat v4.3.0 | *Hao et al., 2021* | RRID:SCR_016341 | |
| Software, algorithm | R package SeuratWrappers v0.3.1 | Satija Lab | RRID:SCR_022555 | |
| Software, algorithm | R package pheatmap v1.0.12 | pheatmap | RRID:SCR_016418 | |
| Software, algorithm | R package velocyto.R v0.6 | *La Manno et al., 2018* | RRID:SCR_018167 | |
| Software, algorithm | Cytoscape v3.10.1 | *Shannon et al., 2003* | RRID:SCR_003032 | |
| Software, algorithm | EnrichmentMap v3.3.6 | *Merico et al., 2010* | RRID:SCR_016052 | |
| Software, algorithm | AutoAnnotate v1.4.1 | *Kucera et al., 2016* | | https://apps.cytoscape.org/apps/autoannotate |
| Software, algorithm | ImageJ v1.51 | NIH | RRID:SCR_003070 | |
| Software, algorithm | GRatio for ImageJ | *Goebbels et al., 2010* | RRID:SCR_015580 | http://gratio.efil.de/ |
| Software, algorithm | Prism v5.01 | GraphPad | RRID:SCR_002798 | |
| Software, algorithm | R v4.2.1 | The R Project for Statistical Computing | RRID:SCR_001905 | |
| Other | DMEM/High glucose | HyClone | Cat# SH30243.01 | Medium used during FACS isolation of cells |
| Other | Horse serum, heat inactivated | Gibco | Cat# 26050–088 | Serum used during FACS isolation of cells |
| Other | Collagenase, type 2 | Worthington Biochemical | Cat# LS004177 | Dissociation enzyme used during FACS isolation of cells |
| Other | Dispase II | Gibco | Cat# 17105–041 | Dissociation enzyme used during FACS isolation of cells |
| Other | HiSeq X Ten | Illumina | Cat# SY-412–1001 | scRNA-seq device |
| Other | TRRUST v2 | *Han et al., 2018* | RRID:SCR_022554 | https://www.grnpedia.org/trrust/ |
| Other | KEGG PATHWAY Database | Kanehisa Laboratories | RRID:SCR_018145 | https://www.genome.jp/kegg/pathway.html |
| Other | g:Profiler | *Kolberg et al., 2023* | RRID:SCR_006809 | https://biit.cs.ut.ee/gprofiler/gost |
| Other | Isolated Pulse Stimulator | A-M Systems | Model 2100 | Electric pulse generator used during CMAP measurement |
| Other | Data Recorder | iWorx | IX-RA-834 | Data recorder used during CMAP measurement |
| Other | Ultramicrotome | Leica | EM UC7 | Used for sciatic nerve semi-thin section generation |
| Other | Light microscope | Thermo Fisher Scientific | EVOS FL Auto 2 | Used for imaging toluidine blue-stained sciatic nerve sections |

## Animals

C57BL/6 J (RRID:IMSR_JAX:000664), *Prrx1^Cre^* (RRID:IMSR_JAX:005584), *Bdnf^fl^* (RRID:IMSR_JAX:004339), *Plp1^CreER^* (RRID:IMSR_JAX:005975), and *Rosa26^LSL-tdTomato^* (RRID:IMSR_JAX:007914) mice were all obtained from The Jackson Laboratory. To generate *Prrx1^Cre^; Bdnf^fl/fl^* mice, *Prrx1^Cre/+^; Bdnf^fl/+^* males; and *Bdnf^fl/fl^* females were crossed to avoid germline recombination in the female reproductive cells, and littermates were used as controls. To generate *Plp1^CreER^; Rosa26^LSL-tdTomato^* mice, *Plp1^CreER/+^* mice were crossed with *Rosa26^LSL-tdTomato/+^* mice, and the line was kept by breeding *Plp1^CreER/+^; Rosa26^LSL-tdTomato/LSL-tdTomato^* males; and females. Only mice that had the *Plp1^CreER^* allele was used. Primers used for genotyping are listed in *Supplementary file 1d*. All mice were bred on the B6 background, except for the *Prrx1^Cre^; Bdnf^fl/fl^* mice, and littermates that were kept in the mixed B6, 129S4, and BALB/c background. All mice were housed in a specific-pathogen-free (SPF) animal facility, with a 12 hr light/12 hr dark cycle at room temperature (RT, 22°C) and 40–60% humidity, and were fed with a normal chow diet and water ad libitum. Tamoxifen administration, sciatic nerve crush injury, denervation, and intramuscular injection of GDNF or GDNF-blocking antibodies were all given to 3–4 month-old adult mice. When comparing aged mice versus adult mice, 24-month-old and 5–6 month-old mice were used, respectively. In all cases except for *Plp1^CreER^; Rosa26^LSL-tdTomato^* mice, male mice were used for the experiments. No sex-specific differences were observed in experiments using *Plp1^CreER^; Rosa26^LSL-tdTomato^* mice. All experimental procedures were approved by the Institutional Animal Care and Use Committee at Seoul National University and were carried out according to the guidelines provided.

## Tamoxifen administration

To label Schwann cells, 3-month-old *Plp1^CreER^; Rosa26^LSL-tdTomato^* mice were administered orally with tamoxifen (20 mg/ml in corn oil, 160 mg/kg body weight; Sigma-Aldrich) three times every other day.

## Sciatic nerve injury

Mice were deeply anesthetized via intraperitoneal injection of Avertin (32 mg/ml, ~800 mg/kg; Sigma-Aldrich), and the incision site on the posterior side of the right hindlimb was shaved and depilated using surgical clippers and hair removal cream. After cleansing the incision site with 70% ethanol, the incision was made on the skin with surgical scissors, and the biceps femoris muscle was punctured open with fine-tip forceps to expose the sciatic nerve. For sciatic nerve crush injury, the exposed nerve was crushed with fine forceps for 30 s at the site just proximal to where the tibial, peroneal, and sural nerves branched out from the sciatic nerve. For denervation, ~5 mm of the sciatic nerve proximal from the crush injury site was cut and removed. The punctured biceps femoris muscle and skin were then sutured, and the incision site was sterilized with povidone-iodine.

## Intramuscular injection of GDNF

Tibialis anterior muscle and the two gastrocnemius muscles (GA, lateral, and medial) were each injected with 10 µl of either PBS or recombinant mouse GDNF (10 µg/ml, Sigma-Aldrich) using 31-gauge insulin syringes without damaging any innervating nerves. The injected muscles were then dissected 48 hr post-injection for isolation of FAPs and further analysis. Mice in the same litter were randomly selected for either PBS or recombinant mouse GDNF injection.

## Intramuscular injection of GDNF-blocking antibodies

Tibialis anterior muscle and the two gastrocnemius muscles (GA, lateral, and medial) were each injected with 10 µg of either normal rabbit IgG (Sino Biological) or anti-GDNF antibodies (Alomone Labs) using 31-gauge insulin syringes 24 hr post-SNC. The injected muscles were then dissected 48 hr post-injection (72 hr post-injury) for isolation of FAPs and further analysis. Mice in the same litter were randomly selected for either IgG or anti-GDNF antibody injection.

## Isolation of FAPs, Schwann cells, and others

Isolation of muscle-resident FAPs, Schwann cells, and others was performed according to a previously reported protocol (*Liu et al., 2015*) with minor modifications. Muscles indicated in each experiment were dissected, finely chopped with surgical scissors, and washed with 10% horse serum (Gibco), and DMEM (HyClone) for further dissociation. Enzymatic dissociation was carried out in 10% horse serum, DMEM containing collagenase II (800 U/ml, Worthington Biochemical) and dispase II (1.1 U/ml,

Gibco) for 40 min at 37°C with mild agitation, and mechanical dissociation was performed by trituration of the dissociated solution with a 20-gauge needle 10 times. After filtering the solution through a 40 µm strainer, dissociated mononuclear single cells were stained with the following antibodies: APC anti-mouse CD31, APC anti-mouse CD45, PE anti-mouse TER-119, biotin anti-mouse CD106 (Vcam1) (Biolegend), and FITC anti-mouse Ly-6A/E (Sca1) (BD Pharmingen). 7-aminoactinomycin D (7-AAD, Sigma-Aldrich) was added to stain dead cells, and PE/Cy7 streptavidin was used to label Vcam1$^+$ cells. Gating strategies used for the isolation of each cell type were as follows: FAPs, 7-AAD$^-$Ter119$^-$CD31$^-$CD45$^-$Vcam1$^-$Sca1$^+$; MuSCs, 7-AAD$^-$Ter119$^-$CD31$^-$CD45$^-$Vcam1$^+$Sca1$^-$; lineage-positive cells, 7-AAD$^-$Ter119$^-$CD31$^+$, or 7-AAD$^-$Ter119$^-$CD45$^+$; double-negative cells, 7-AAD$^-$Ter119$^-$CD31$^-$CD45$^-$Vcam1$^-$Sca1$^-$. For Schwann cells, 7-AAD$^-$tdTomato$^+$ cells were sorted from tamoxifen-administered *Plp1$^{CreER}$; Rosa26$^{LSL-tdTomato}$ Rosa26$^{LSL-tdTomato}$* mice.

## scRNA-seq library construction and sequencing

FAPs were isolated from sciatic nerve crush injury-affected or denervated muscles on days 3, 7, 14, and 28 post-injury using wild-type B6 mice, so that a total of nine samples, including uninjured control, were collected for library generation. For each sample, isolated FAPs were pooled from two mice. Chromium Next GEM Single Cell 3′ Kit v3.1 (10x Genomics) was used according to the manufacturer's instructions for the nine collected FAP samples, and the target cell number for recovery was set to 5000 in each sample. Sequencing of the libraries was carried out using HiSeq X Ten (Illumina).

## Computational analysis of scRNA-seq data

Sequenced reads were aligned to the mouse reference genome mm10 using CellRanger v3.1.0 (10x Genomics), and aligned reads were transformed into gene-cell count matrices using velocyto v0.17 (*La Manno et al., 2018*) to obtain count matrices for both spliced and unspliced mRNAs. Output loom files were then loaded with R package SeuratWrappers v0.3.1, and were preprocessed and analyzed using R package Seurat v4.3.0 (*Hao et al., 2021*) for downstream analysis. The preprocessing steps for quality control included doublet filtering, live cell filtering, and removal of non-FAPs as previously described (*Kim et al., 2022*). All nine sample data were merged and normalized for dimensionality reduction, where the top eight principal components from the principal component analysis using 5000 variable genes were selected for two-dimensional UMAP embedding and visualization. Unsupervised clustering of cells was achieved through FindNeighbors and FindClusters functions in the Seurat R package. To identify DEGs in the pairwise comparisons of scRNA-seq samples, the FindMarkers function in Seurat R package was used with the following parameters: fold change ≥ 2, pseudocount.use=0.01, min.pct=0.01, adjusted *p*-value <0.05. For identification of DEGs in each cluster, the FindAllMarkers function was used with the parameters: fold change ≥ 1.5, pseudocount.use=0.01, min.pct=0.02, adjusted *p*-value <0.05. For hierarchical clustering, the R package pheatmap v1.0.12 was used. For RNA velocity analysis, R package velocyto.R v0.6 was used following the instructions provided by the developer (*La Manno et al., 2018*). For the prediction of upstream regulatory transcription factors using lists of DEGs enriched in the selected FAP clusters, the web-based tool TRRUST v2 (*Han et al., 2018*) was used. For color mapping of the MAPK signaling pathway, the pathway image from the KEGG PATHWAY Database (*Kanehisa et al., 2023*) was retrieved and colored manually.

## Gene set overrepresentation analysis

To identify pathways enriched in the lists of DEGs, a web-based version of g:Profiler (*Kolberg et al., 2023*) was used with the following parameters: organism – *Mus musculus*; ordered query – YES; data sources – GO biological process without electronic GO annotations, Reactome, and WikiPathways; advanced options were set to default. Visualization of the results were done as previously described (*Reimand et al., 2019*), using the Cytoscape v3.10.1 (*Shannon et al., 2003*) application with tools EnrichmentMap v3.3.6 (*Merico et al., 2010*) and AutoAnnotate v1.4.1 (*Kucera et al., 2016*).

## RNA extraction and qRT-PCR

Total RNA extraction and reverse transcription was carried out for isolated FAPs, Schwann cells, MuSCs, and others using TRIzol reagent (Invitrogen) and ReverTra Ace qPCR RT Master Mix (Toyobo) reagents respectively, following the manufacturer's instructions. qPCR was performed using ORA SEE qPCR Green ROX L Mix (HighQu) reagent, with gene-specific primers listed in *Supplementary file*

1d. Quantitative analysis of mRNA levels were done using the $2^{-\Delta\Delta Ct}$ method with β-actin (*Actb*) as the housekeeping gene for normalization.

## Western blot analysis

FACS-isolated FAPs were lysed in RIPA buffer (Biosesang) added with 1 x Halt protease inhibitor cocktail (Thermo Scientific), and were sonicated. The lysate was then centrifuged at 13,000 rpm for 30 min at 4°C, and the supernatant was mixed with Laemmli sample buffer before heating at 95°C for 10 min. Samples were subjected to electrophoresis in 10% polyacrylamide gels and transferred to 0.45 μm PVDF membranes (Immobilon). For GAPDH (rabbit anti-GAPDH, 1:1000, Bethyl Laboratories) and PDGFRα (rabbit anti-PDGFRα, 1:200, Santa Cruz Biotechnology), membranes were blocked in 5% skim milk (LPS Solution), 0.1% Tween-20 (Sigma-Aldrich) in TBS, and incubated with primary antibodies overnight at 4°C, and then with the secondary antibody (horseradish peroxidase-conjugated anti-rabbit IgG, 1:10,000, Promega) for 1 hr at RT. For BDNF (rabbit anti-BDNF, 1:1000, Alomone Labs), membranes were blocked in 5% bovine serum albumin (Bovogen Biologicals), 0.1% Tween-20 in TBS, and incubated with the primary antibody in immunoreaction enhancer solution 1 (Toyobo) for 1 hr at RT, and then with the same secondary antibody in immunoreaction enhancer solution 2 (Toyobo) for 1 hr at RT. The membranes were then developed using SuperSignal West Dura Extended Duration Substrate (Thermo Scientific) according to the manufacturer's instructions, and imaged with FUSION Solo chemiluminescence imaging system (Vilber Lourmat). Densitometric quantification of the imaged data were performed using ImageJ v1.51n (NIH).

## Electromyography and CMAP measurement

Intraperitoneal injection of Avertin (32 mg/ml,~800 mg/kg) for anesthetization of mice was carried out prior to CMAP measurement. Stimulation of the sciatic nerve was achieved by placing stimulating electrodes subcutaneously on either side of the sciatic notch and applying supramaximal stimuli (~70 mA) at a rate of 1 pulse per second with a duration of 0.1 ms, using Isolated Pulse Stimulator Model 2100 (A-M Systems). Recording electrode was placed carefully on the GA muscle subdermally without puncturing the muscle, with the reference electrode placed near the Achilles tendon and the ground electrode placed on the tail for data recording using Data Recorder IX-RA-834 (iWorx). CMAP amplitude was determined by the absolute difference between potentials of positive and negative peaks, and CMAP latency was determined by the delay from stimulus peak to the beginning of response peak. Three individual measurements were taken from each animal's GA muscle, and average values were used as representatives for statistical analysis. The measurements were obtained by a single evaluator (K. Y.) blinded to the genotype.

## Toluidine blue staining of sciatic nerves

Sciatic nerve distal to the injury site was dissected at 6 weeks post-injury for analysis. The dissected nerves were fixed in 4% paraformaldehyde dissolved in Sorensen's phosphate buffer (0.1 M, pH 7.2) at 4°C overnight, followed by procedures described previously (*Kim et al., 2022*) with minor modifications for semi-thin sectioning. Briefly, fixed samples were washed with 0.1 M sodium cacodylate buffer (pH 7.2), post-fixed with 1% osmium tetroxide in 0.1 M sodium cacodylate buffer (pH 7.2) for 1 hr at RT, washed with distilled water (DW) and stained with 0.5% uranyl acetate at 4°C overnight. Stained samples were then washed with DW and dehydrated using serial ethanol and propylene oxide. Samples were then embedded in Spurr's resin (Electron Microscopy Sciences), and semi-thin sections (500 nm) were prepared with a diamond knife on an ultramicrotome EM UC7 (Leica). The sections were dried down on glass slides for staining and light microscopy. Toluidine blue staining were done using 1% toluidine blue solution containing 1% sodium borate on a slide warmer (70°C), and images were obtained with the light microscope EVOS FL Auto 2 (Thermo Fisher Scientific) for analysis.

## G-ratio quantification

Semi-automated quantification of myelinated axon diameters were carried out using an ImageJ plugin for g-ratio quantification (*Goebbels et al., 2010*), where axon diameters and G-ratios were quantified and calculated, respectively. G-ratios were calculated as [naked axon diameter]/[myelinated axon

diameter]. Selection of axons for quantification was randomized by the ImageJ plugin, and a single evaluator (K. Y.) performed the semi-automated quantification blinded to the genotype.

## Quantification and statistical analysis

All statistical analyses were performed using Prism v5.01 (GraphPad) and R v4.2.1. Continuous variables were tested for normal distribution with the Shapiro-Wilk test, and the F-test was used to check for equal variance. For comparison of significant differences in multiple groups, a one-way analysis of variance (ANOVA) followed by Bonferroni's pairwise post hoc test was applied. For the comparison of the two groups, an unpaired t-test was used for data with normal distribution and equal variance; Welch's t-test was used for data with normal distribution and unequal variance; and the Mann-Whitney U test was used for non-normally distributed data. ANCOVA was applied to test for differences between slopes of linear regression lines. Two-way ANOVA was applied to compare two groups with two variables. For comparison of gene expression levels between scRNA-seq data, the Wilcoxon rank sum test was applied as a default in functions FindAllMarkers and FindMarkers within the R package Seurat v4.3.0. For RT-qPCR, the average of triplicate technical values were used for each biological replicate. All error bars represent mean ± SD. p-value of less than 0.05 was considered statistically significant at the 95% confidence level. The number of technical and biological replicates and statistical analyses used in each experiment are indicated in the figure legends.

## Acknowledgements

We thank Dr. Jong-Eun Park for his helpful comments on the analysis of our scRNA-seq data. *Prrx1*[Cre] mice, *Bdnf*[fl] mice, and *Plp1*[CreER] mice were obtained via the Korea Mouse Phenotyping Center, deposited by Dr. Rho Hyun Seong, Dr. Yun-Hee Lee, and Dr. Myunghwan Choi, respectively. Also, we would like to acknowledge technical support from the Center for Research Facilities, Biological Sciences Department, Seoul National University (SNU), and SNU National Instrumentation Center for Environmental Management. Library construction and sequencing of the scRNA-seq data was performed by Macrogen Inc Experimental schemes and graphical summaries in this paper were created using BioRender. This work was supported by grants from the National Research Foundation of Korea (NRF-2020R1A5A1018081, NRF-2022R1A2C3007621) funded by the Korean government (MSIT).

## Additional information

### Funding

| Funder | Grant reference number | Author |
| --- | --- | --- |
| National Research Foundation of Korea | NRF-2020R1A5A1018081 | Kyusang Yoo<br>Young-Woo Jo<br>Sang-Hyeon Hann<br>Inkuk Park<br>Yea-Eun Kim<br>Ye Lynne Kim<br>Joonwoo Rhee<br>In-Wook Song<br>Young-Yun Kong |
| National Research Foundation of Korea | NRF-2022R1A2C3007621 | Kyusang Yoo<br>Young-Woo Jo<br>Sang-Hyeon Hann<br>Inkuk Park<br>Yea-Eun Kim<br>Ye Lynne Kim<br>Joonwoo Rhee<br>In-Wook Song<br>Young-Yun Kong |

The funders had no role in study design, data collection and interpretation, or the decision to submit the work for publication.

## Author contributions
Kyusang Yoo, Conceptualization, Resources, Data curation, Software, Formal analysis, Validation, Investigation, Visualization, Methodology, Writing - original draft, Writing – review and editing; Young-Woo Jo, Resources, Formal analysis, Validation, Investigation, Visualization, Writing – review and editing; Takwon Yoo, Software, Validation; Sang-Hyeon Hann, Inkuk Park, Joonwoo Rhee, In-Wook Song, Ji-Hoon Kim, Resources; Yea-Eun Kim, Resources, Writing – review and editing; Ye Lynne Kim, Writing – review and editing; Daehyun Baek, Validation; Young-Yun Kong, Conceptualization, Supervision, Funding acquisition, Validation, Project administration, Writing – review and editing

## Author ORCIDs
Kyusang Yoo ⓘ https://orcid.org/0000-0002-9852-5344
Takwon Yoo ⓘ https://orcid.org/0009-0000-4381-7553
Sang-Hyeon Hann ⓘ https://orcid.org/0000-0002-0871-3414
Young-Yun Kong ⓘ https://orcid.org/0000-0001-7335-3729

## Ethics
All experimental procedures were approved by the Institutional Animal Care and Use Committtee at Seoul National University (Permit Numbers: SNU-200509-2-2, SNU-201202-1-3, SNU-210118-5-1, SNU-230415-1-1, SNU-231218-1) and were carried out according to the guidelines provided.

Reviewer #1 (Public Review): https://doi.org/10.7554/eLife.97662.3.sa1
Author response https://doi.org/10.7554/eLife.97662.3.sa2

# Additional files

## Supplementary files
• Supplementary file 1. Supplemental tables showing results from TRRUST analyses, candidate effector gene categorization, and primer sequences used in this study. (a) Results from TRRUST show transcription factors predicted to regulate genes specifically enriched in cluster 2. Genes known to be regulated by each transcription factor is listed. (b) Results from TRRUST showing transcription factors predicted to regulate genes specifically enriched in cluster 3. Genes known to be regulated by each transcription factor is listed. (c) Categorization of genes predicted to be regulated by transcription factors that act downstream of the glial cell line-derived neurotrophic factor (GDNF) signaling pathway. Note that only *Bdnf* fits into all three criteria. (d) Primers used in this study for genotyping PCR or RT-qPCR.
• MDAR checklist

## Data availability
Single-cell RNA-sequencing data produced in this study have been deposited at GEO under accession number GSE250436. Study design, data analysis, and reporting the data were done according to the ARRIVE guidelines (*Kilkenny et al., 2010*).

The following dataset was generated:

| Author(s) | Year | Dataset title | Dataset URL | Database and Identifier |
|---|---|---|---|---|
| Yoo K, Kong Y-Y | 2024 | Muscle-resident mesenchymal progenitors sense and repair peripheral nerve injury via the GDNF-BDNF axis | https://www.ncbi.nlm.nih.gov/geo/query/acc.cgi?acc=GSE250436 | NCBI Gene Expression Omnibus, GSE250436 |

The following previously published datasets were used:

| Author(s) | Year | Dataset title | Dataset URL | Database and Identifier |
|---|---|---|---|---|
| Carr MJ, Toma JS, Johnston AP, Steadman PE, Yuzwa SA, Mahmud N, Frankland PW, Kaplan DR, Miller FD | 2019 | Mesenchymal precursor cells in adult nerves contribute to mammalian tissue repair and regeneration | https://www.ncbi.nlm.nih.gov/geo/query/acc.cgi?acc=GSE120678 | NCBI Gene Expression Omnibus, GSE120678 |
| Toma JS, Karamboulas K, Carr MJ, Kolaj A, Yuzwa SA, Mahmud N, Storer MA, Kaplan DR, Miller FD | 2020 | Peripheral Nerve Single-Cell Analysis Identifies Mesenchymal Ligands that Promote Axonal Growth (scRNAseq data) | https://www.ncbi.nlm.nih.gov/geo/query/acc.cgi?acc=GSE147285 | NCBI Gene Expression Omnibus, GSE147285 |
| Giger RJ, Johnson CN, Zhao X | 2022 | Injured Sciatic Nerve Atlas (iSNAT) | https://www.ncbi.nlm.nih.gov/geo/query/acc.cgi?acc=GSE198582 | NCBI Gene Expression Omnibus, GSE198582 |
| Puri PL, Nicoletti C, Wei X | 2023 | Muscle denervation promotes functional interactions between glial and mesenchymal cells through NGFR and NGF | https://www.ncbi.nlm.nih.gov/geo/query/acc.cgi?acc=GSE221736 | NCBI Gene Expression Omnibus, GSE221736 |

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
