## [Editor Report · eLife assessment]

The study has identified a cell type in muscle that is characterized as an adipogenic progenitor cell that is capable of promoting regeneration through the action of BDNF, a prominent growth factor regulated by GDNF in Schwann cells. These results represent an **important** cellular explanation for nerve regeneration. The revised analysis is **solid** but the work remains **incomplete** due to a lack of evidence that BDNF is produced during the process through the action of GDNF.

---

## [Referee Report · Reviewer #1 (Public Review)]

In this manuscript, Yoo et al describe the role of a specialized cell type found in muscle, Fibro-adipogenic progenitors (FAPs), in promoting regeneration following sciatic nerve injury. Using single-cell transcriptomics, they characterize the expression profiles of FAPs at various times after nerve crush or denervation. Their results reveal that a population of these muscle-resident mesenchymal progenitors up regulate the receptors for GDNF, which is secreted by Schwann cells following crush injury, suggesting that FAPs respond to this growth factor. They also find that FAPs increase expression of BDNF, which promotes nerve regeneration. The authors demonstrate FAP production of BDNF in vivo is up regulated in response to injection of GDNF and that conditional deletion of BDNF in FAPs results in delayed nerve regeneration after crush injury, primarily due to lagging remyelination. Finally, they also find reduced BDNF expression following crush injury in aged mice, suggesting a potential mechanism to explain the decrease in peripheral nerve regenerative capability in aged animals. These results are very interesting and novel and provide important insights into the mechanisms regulating peripheral nerve regeneration, which has important clinical implications for understanding and treating nerve injuries.

However, the authors should provide more compelling evidence that BDNF is produced by FAPs in response to GDNF signaling. The suggestion that Schwann cell-derived GDNF is responsible for up regulation of BDNF in the FAPs is primarily indirect, based on the data showing that injection of GDNF into the muscle is sufficient to up regulate BDNF (Fig. 4H). The authors more directly test their hypothesis by administering GDNF blocking antibody and find a trend toward reduced BDNF (Fig. 4S2), but it is not statistically significant at this point. Additional replicates should be performed to determine if BDNF levels are indeed reduced when GDNF is blocked.

---

## [Author Response]

The following is the authors’ response to the original reviews.

Point-by-point reply in response to the Reviewer’s comments

**Reviewer #1**

**Public review:**
[1] (a) Given that only a fraction of the FAPs express BDNF after injury, the authors need to demonstrate the specificity of the Prrx1-Cre for FAPs. This is particularly important because muscle stem cell also express GDNF receptors (Fig. 3C & D) and myogenic progenitors/satellite cells produce BDNF after nerve injury (Griesbeck et al., 1995 (PMID 8531223); Omura et al., 2005 (PMID 16221288)). (b) Moreover, as the authors point out, there are multipotent mesenchymal precursor cells in the nerve that migrate into the surrounding tissue following nerve injury and contribute to regeneration (Carr et al, PMID 30503141). Therefore, there are multiple possible sources of BDNF, highlighting the need to clearly demonstrate that FAP-derived BDNF is essential.

- (a) As the Reviewer noted, both GDNF receptor expression and increased BDNF expression in response to nerve injury are detectable in both FAPs and muscle stem cells (MuSCs). Therefore, we agree with the Reviewer that demonstrating the specificity of Prrx1-Cre in FAPs is crucial to support our claim. In our previous publication (Kim et al., 2022), using Prrx1-Cre; Rosa-eYFP mice, we showed that while most of the CD31-CD45-Vcam1-Sca1+ FAPs are eYFP+, CD31-CD45-Vcam1+Sca1- MuSCs do not express eYFP (Liu et al., 2015; Kim et al., 2022) (Attached Figure 1). Additionally, genomic DNA PCR using mononuclear cells sorted from our *Prrx1Cre*; *Bdnffl/fl* mice showed that DNA recombination in the floxed *Bdnf* gene could only be detected in FAPs and CD31-CD45-Vcam1-Sca1- cells, but not in MuSCs (Author response image 2). This is consistent with a previous report that showed Prrx1-Cre activity in FAPs, pericytes, vascular smooth muscle cells (vSMCs) and tenocytes (Leinroth et al., 2022), where pericytes, vSMCs and tenocytes are included the CD31-CD45-Vcam1Sca1- population (Giordani et al., 2019). Together, these results demonstrate that while Prrx1-Cre is active in FAPs, it is absent in MuSCs.

**Author response image 1. sa2fig1:** Expression of eYFP in muscle-resident, lineage-negative, live mononuclear cells isolated from *Prrx1Cre*;*RosaeYFP* mice. Supplemental Figure 3A from Kim et al., 2022. Lin-: lineage-negative (CD31-CD45-); Neg.: Vcam1-Sca1-.

**Author response image 2. sa2fig2:** Recombination of the floxed *Bdnf* gene in the mononuclear cells sorted from muscles of *Prrx1Cre*; *Bdnffl/fl* or *Bdnffl/fl* mice. Genotypes and cell types sampled for each lane is specified. P4, P5, and P6 indicate primers used for each PCR. Lin+: lineage(CD31/CD45)-positive; DN: CD31-CD45-Vcam1-Sca1-.

- (b) We appreciate and agree with the Reviewer’s comment that additional experiments are needed to confirm that FAP-derived BDNF is indeed essential for nerve regeneration, considering other potential cellular sources of BDNF, such as nerve-resident mesenchymal precursor cells. One possible experiment that could demonstrate the requirement of FAP-derived BDNF in nerve regeneration would be the transplantation of wild-type FAPs into our *Prrx1Cre*; *Bdnf fl/fl* mice and to see if the delay in nerve regeneration and remyelination is recovered, making the process similar to that in control mice. Unfortunately, since the genetic background of our *Prrx1Cre*; *Bdnffl/fl* mice is a mixture of B6, 129S4, and BALB/c, immune rejection of the transplanted cells may occur, which makes the experiment technically difficult. Another experimental approach could involve the use of FAP-specific Cre mouse line, as we have mentioned in the Discussion of our original manuscript. However, such a line does not yet exist due to the lack of a marker gene that is expressed specifically in FAPs, but not in nerve-resident mesenchymal precursor cells. Overcoming such technical challenges and demonstrating the requirement of FAP-derived BDNF in nerve regeneration would significantly strengthen our report, though we regret that these methods are currently unavailable.

[2] Similarly, the authors should provide some evidence that BDNF protein is produced by FAPs. All of their data for BDNF expression is based on mRNA expression and that appears to only be increased in a small subset of FAPs. Perhaps an immunostaining could be done to demonstrate up-regulation of BDNF in FAPs after injury.

- We appreciate the Reviewer’s constructive comment. To demonstrate that BDNF protein is produced by FAPs upon nerve injury, we performed western blot analysis. FAPs were isolated from either sciatic nerve crush injury-affected muscles at 7 days post injury (dpi) or from the contralateral, uninjured muscles, and protein samples were prepared for SDS-PAGE and western blot using anti-BDNF, anti-PDGFRα and antiGAPDH antibodies. As a result, while both nerve injury-affected and uninjured musclederived FAPs expressed PDGFRα, the mature from of BDNF protein was only detected in nerve injury-affected FAPs, showing that BDNF is indeed expressed in FAPs at the protein level after injury. We have added this new result as Figure 4F in the New Figure 4 with the experimental scheme as New Figure 4—figure supplement 1, and revised the Results section (lines 364-374) and the Materials and Methods section (lines 687-705) in our manuscript to include the new results in detail.

[3] The suggestion that Schwann cell-derived GDNF is responsible for upregulation of BDNF in the FAPs is indirect, based largely on the data showing that injection of GDNF into the muscle is sufficient to up-regulate BDNF (Fig. 4F & G). However, to more directly connect the 2 observations in a causal way, the authors should inject a Ret/GDNF antagonist, such as a Ret-Fc construct, then measure the BDNF levels.

- We appreciate the Reviewer’s constructive comment, and we agree that testing the necessity of GDNF/RET signaling in BDNF upregulation is crucial to link the expression of the two neurotrophic factors in a causal way. As a means to antagonize GDNF/RET signaling, we injected anti-GDNF antibodies into the tibialis anterior and gastrocnemius muscles following sciatic nerve crush injury to block the activity of intramuscular GDNF protein. As a result, although the differences were not statistically significant, we observed a tendancy towards decreased *Bdnf* mRNA expression upon anti-GDNF injection compared to IgG controls. We have added this new result as New Figure 4—figure supplement 2, and revised our manuscript to include the details in both the Results section (lines 381-390) and the Materials and Methods section (lines 611-616). We have also changed the title of New Figure 4 (line 332) to encompass the new results. We are aware that further experiments that may involve increasing the number of animals tested, increasing the antibody injection dosage or frequency, or implementation of genetic models such as *Plp1CreER*; *Gdnffl/fl* should be carried out to validate our hypothesis with statistical significance. Unfortunately, due to limited time, resources, and research funds, we were unable to perform such additional experiments. We hope that the Reviewer understands these limitations.

[4] (a) In assessing the regeneration after nerve crush, the authors focus on remyelination, for example, assessing CMAP and g-ratios. However, they should also quantify axon regeneration, which can be done distal to the crush injury at earlier time points, before the 6 weeks scored in their study. Evaluating axon regeneration, which occurs prior to remyelination, would be especially useful because BDNF can act on both Schwann cells, to promote myelination, and axons, enhancing survival and growth. (b) They could also evaluate the stability of the neuromuscular junctions, particularly if a denervation was done with the conditional knock outs, although that may be a bit beyond the scope of this study.

- (a) As the Reviewer mentioned, BDNF is known to act on both Schwann cells and axons, where it promotes myelination and axonal growth, respectively (Oudega and Hagg, 1998; Zhang et al., 2000; Chan et al., 2001; Xiao et al., 2009; English et al., 2013). We fully agree with the Reviewer’s comment that quantification of axon regeneration, which could be achieved through immunostaining of the distal part of the sciatic nerve at earlier time points after injury, would shed light on whether FAPderived BDNF can also contribute to axon regeneration in addition to remyelination. Unfortunately, we could not perform such additional experiments within the limited time frame, since preparing enough numbers of control and conditional knockout mice that match the age groups used in this study (3-4 months old), followed by waiting for additional 2-4 weeks after nerve crush injury for sample collection, and subsequent immunostaining for quantification could take almost 6 months in total. We hope that the Reviewer understands this limitation.

- (b) We appreciate the Reviewer’s constructive comment. Although the number of animals used for neuromuscular junction (NMJ) analyses was not sufficient, we had briefly examined the structure of NMJs at 4 weeks post nerve crush injury in control (Ctrl) and conditional knockout (cKO) mice as a preliminary experiment. As a result, no significant differences were observed between Ctrl and cKO mice in terms of NMJ morphology and innervation (Author response image 3).

**Author response image 3. sa2fig3:** Structures of neuromuscular junctions from Ctrl vs cKO mice at 4 weeks post nerve crush injury. Whole-mount immunostaining was done using the exterior digitorum longus muscles that were affected by sciatic nerve crush injury. Samples were stained with α-bungarotoxin (green), neurofilament (red), and synaptophysin (blue). Scale bar: 50 μm.

Going back to part (a) of this Reviewer’s comment, considering the data presented in Author response image 3, where innervation of axons into acetylcholine receptor clusters was not significantly different between Ctrl versus cKO mice, FAP-derived BDNF may not be critical for the axonal growth upon nerve injury. Although we acknowledge that additional experiments are required to draw a meaningful conclusion on this point, we could not perform such additional experiments due to insufficient time and resources.

We hope that the Reviewer understands our limitation.

**Recommendations for the authors:**
[1] In citing the ability of BDNF to promote Schwann cell myelination the authors should include Chan et al., 2001 (PMID 11717413) in addition to the Zhang et al, 2000 and Xiao et al, 2009 references.

- We apologize for missing out the reference mentioned by the Reviewer. We have added the suggested reference in our revised manuscript (lines 395, 425, and 517).

**Reviewer #2**

**Public review:**
[1] Although, I find the data the authors generated enough for their claims. I do see them as relatively poor, and (a) a complementary analysis of protein expression would strengthen the paper through immunostaining of the different genes mentioned for FAPs and Schwann cells. The model is entirely supported by measuring mRNA levels and negative regulation of gene expression in specific cells. Additionally, (b) what happens to the structure of the neuromuscular junction after regeneration when GDNF or BDNF expression is reduced? (c) The determination of decreasing levels of FAPs BDNF mRNA during aging is interesting; is the gain of BDNF expression in FAPs reverting the phenotype?

- (a) We appreciate and agree with the Reviewer’s comment that validation of BDNF protein expression in FAPs and GDNF protein expression in Schwann cells upon nerve injury would strengthen this paper. Regarding GDNF protein expression in Schwann cells upon nerve injury, it has already been demonstrated by previous studies (Höke et al., 2002; Xu et al., 2013). For BDNF protein expression in FAPs upon nerve injury, we performed western blot analysis for validation, as mentioned in the response to Reviewer #1 Public review [2]. The results showed that while the mature form of BDNF protein could not be readily detected in FAPs isolated from uninjured muscles, it could be detected in FAPs isolated from sciatic nerve crush injury-affected muscles at 7 days post injury. We have added the new result as Figure 4F in the New Figure 4 with the experimental scheme as New Figure 4—figure supplement 1, and revised the Results section (lines 364-374) and the Materials and Methods section (lines 687-705) in our manuscript to include the new results in detail.

- (b) Though the data is preliminary, we examined the structures of neuromuscular junctions (NMJs) from control and *Prrx1Cre*; *Bdnf fl/fl* mice at 4 weeks post injury in the exterior digitorum longus muscles, as mentioned in the response to Reviewer #1 Publilc review [4](b). As a result, we could not identify significant differences between control versus *Prrx1Cre*; *Bdnf fl/fl* mice, where BDNF expression is reduced specifically in *Prrx1*-expressing cells, including FAPs (Attached Figure 3). Since other cellular sources of BDNF, such as Schwann cells, exist, regeneration of the NMJs may not have been as significantly affected as remyelination in our *Prrx1Cre*; *Bdnf fl/fl* mice. However, further experiments with a sufficient number of mice and more observation time points are required to statistically validate this hypothesis in detail. Unfortunately, preparing samples for such additional analyses would take more than four months, as we need to produce sufficient numbers of control and *Prrx1Cre*; *Bdnf fl/fl* mice that match the age groups used in this study. We hope that the Reviewer understands our limitation.

Regarding analyzing NMJ structures after regeneration affected by reduced GDNF levels, using genetic models such as *Plp1CreER*; *Gdnffl/fl* mice would be appropriate, as we have used the *Prrx1Cre*; *Bdnffl/fl* mice in this study to reduce BDNF levels produced by FAPs. Unfortunately, we do not have the *Gdnffl* mice, and obtaining these mice to produce *Plp1CreER*; *Gdnffl/fl* mice and performing the additional experiment would take too much time for this current revision. In a further study, we will try to perform the additional experiment by obtaining the required mouse line. We hope that the Reviewer understands our limitation.

- (c) We appreciate the Reviewer for highlighting this point. In this paper, we have shown that BDNF expression upon nerve injury is decreased in aged FAPs compared to young adult FAPs, and suggested that this may be one of the causes of the delayed nerve regeneration phenotype in aged mice. Previously, it has been reported that while intramuscular injection of BDNF accelerates nerve regeneration, intramuscular injection of anti-BDNF antibodies delays the regeneration process (Zheng et al., 2016). This implies that intramuscular levels of active BDNF can significantly influence the speed of nerve regeneration. Therefore, the gain of BDNF expression in aged FAPs may contribute to reversing the delayed nerve regeneration phenotype in aged mice, since it would result in additional supply of active, intramuscular BDNF, which has previously been shown to accelerate nerve regeneration. Though experimental validation is required to support such claim, we could not obtain sufficient numbers of aged mice within the limited time frame. We hope that the Reviewer understands our limitation.

**Recommendations for the authors:**
[1] The authors should include the experimental design and several drawings in the leading figures indicating, for example, how remyelination after injury was quantified and how the response of regenerated sciatic nerve to a depolarizing stimulus was studied.

- We apologize for any confusion caused by insufficient information provided in the leading figures. Unfortunately, due to limited space, we could not add experimental designs or drawings in the leading figures. Instead, to do our best to comply with the

Reviewer’s comment, we have revised the figure legends in the leading figures so that the experimental designs or diagrams can be referred to in the figure supplements.

We hope that the Reviewer understands this limitation.

**Reviewer #3**
Public review:[1] In Fig. 1 and 2 authors provide data on scRNA seq and this is important information reporting the finding of RET and GFRa1 transcripts in the subpopulation of FAP cells. However, authors provide no data on the expression of RET and GFRa1 proteins in FAP cells.

- Reply for this comment by the Reviewer is in the Recommendations for the authors section below ([2]), as the same comment is repeated.

[2] Another problem is the lack of information showing that GDNF secreted by Schwann cells can activate RET and its down-stream signaling in FAP cells. There is no direct experimental proof that GDNF activating GFRa1-RET signaling triggers BDNF upregulation In FAP cells. The data that GDNF signaling is inducing the synthesis and secretion of BDNF is also not conclusive.

- Reply for this comment by the Reviewer is in the Recommendations for the authors section below ([3]), as the same comment is repeated.

Recommendations for the authors:[1] Although this is a novel study and contains very well-performed parts, the GDNF section is preliminary and requires additional experimentation. In the introduction authors describe well FAPs but even do not mention how GDNF is signaling. Moreover, the reader may get an impression that Ras-MAPK pathway is the only or at least the main GDNF signaling pathway. In fact, for neurons Akt and Src signaling pathways play also crucial role.

- We apologize for the missing content in the Introduction section of our manuscript and for any confusion caused by our misleading description of the GDNF signaling pathway. We have revised our manuscript to include the GDNF signaling pathway in the Introduction section, along with a description of other downstream signaling pathways of GDNF that are known to play crucial roles, as mentioned by the Reviewer (lines 115-130). Additionally, we changed the expression in the Results section to avoid making any misleading impressions (lines 318-319).

[2] In Fig. 1 and 2 authors provide data on scRNA seq and this is important information reporting the finding of RET and GFRa1 transcripts in the subpopulation of FAP cells. However, authors provide no data on the expression of RET and GFRa1 proteins in FAP cells.

- We appreciate the Reviewer for the constructive comment. Though we fully agree with the Reviewer that validating the expression of RET and GFRα1 proteins in FAPs is needed, we were unable to obtain the antibodies required for such experiments within the limited time frame for this revision. We hope that the Reviewer understands our limitation. Although we could not directly show the expression of those GDNF receptor genes at the protein level in FAPs, based on the result where intramuscular GDNF injection could sufficiently induce *Bdnf* expression in FAPs compared to PBS control in the absence of nerve damage, it is likely that GDNF receptors are indeed expressed at the protein level in FAPs, since if otherwise, FAPs would not have been able to respond to the injected GDNF protein. Nevertheless, in a future study, we will try to validate the protein-level expression of GDNF receptors in FAPs to comply with the Reviewer’s suggestion and to further support this study.

[3] Another problem is the lack of information showing that GDNF secreted by Schwann cells can activate RET and its down-stream signaling in FAP cells. Authors can monitor activation of MAPK pathway by detecting phospho-Erk and PI3 kinase-Akt pathway measuring phospho-S6 using immunohistochemistry. We can recommend to use the following antibodies: pErk1/2 (1:300, Cell Signaling, Cat# 4370L RRID:AB_2297462), pS6 (1:300, Cell Signaling, Cat# 4858L RRID:AB_1031194). These experiments are crucial because RET and GFRa1 proteins maybe not expressed at the sufficient level on the cell surface.

- We sincerely appreciate the Reviewer’s constructive comment. In this study, we suggested that the GDNF-BDNF axis within FAPs would signal through the MAPK pathway based on the bioinformatic analysis of our single cell RNA-seq data and matching the results with the previously known pathways. We fully agree that monitoring the activation of the MAPK pathway and the PI3K-Akt pathway by immunohistochemistry would experimentally demostrate whether GDNF can activate those pathways within FAPs through GFRα1/RET activation. Unfortunately, we could not obtain the antibodies suggested by the Reviewer for this revision due to insufficient research funds and limited time frame. We hope that the Reviewer understands our limitation. In future studies, we will try to validate the detailed molecular pathway that mediates the GDNF-BDNF axis in FAPs by incorporating the methodology suggested by the Reviewer, along with implementation of genetic models such as *Plp1CreER*; *Gdnffl/fl*, *Prrx1Cre*; *Retfl/fl* or *Prrx1Cre*; *Gfra1fl/fl* to validate whether Schwann cell-derived GDNF can actually signal through its canonical receptor RET/GFRα1 expressed in FAPs to induce expression of BDNF upon nerve injury.

[4] (a) There is no direct experimental proof that GDNF activating GFRa1-RET signaling triggers BDNF upregulation in FAP cells. Authors can use GDNF blocking antibodies, siRNA or use RET or GFRa1 cKO mice to delete them from FAP cells. (b) The data that GDNF signaling is inducing the synthesis and secretion of BDNF is also not conclusive. Authors should show that GDNF injection is increasing BDNF protein levels in FAPs. To get sufficient material for ELISA detection of BDNF is perhaps problematic. However, authors can use BDNF antibodies from Icosagen company and use IHC.

- (a) We appreciate the Reviewer for the critical comment. As mentioned in the reply for Reviewer #1 Public review [3], we used GDNF blocking antibodies to reduce GDNF signaling within the tibialis anterior and gastrocnemius muscles by intramuscular injection after sciatic nerve crush injury, and included the result as a new figure supplement in our revised manuscript (New Figure 4—figure supplement 2) with its details in both the Results section (lines 381-390) and the Materials and Methods section (lines 611-616). Though the results were not statistically significant, intramuscular injection of anti-GDNF antibodies showed a tendency toward reduced *Bdnf* expression in FAPs, compared to IgG controls. As mentioned in the reply for Reviewer #1 Public review [3], and as suggested by the Reviewer, using cKO mice such as *Plp1CreER*; *Gdnffl/fl*, *Prrx1Cre*; *Retfl/fl*, or *Prrx1Cre*; *Gfra1fl/fl* mice would further validate the GDNF-BDNF axis suggested in this study, likely with statistical significance. Unfortunately, obtaining these genetic models within the limited time frame of this current revision is not feasible. We will try to adopt such models in our future study to validate the role of Schwann cell-derived GDNF in inducing BDNF expression in FAPs via activation of RET/GFRα1.

- (b) We appreciate the Reviewer for the constructive comment. Though we fully agree that the experiment suggested by the Reviewer would validate the synthesis and secretion of BDNF protein by GDNF signaling in FAPs, we were not able to perform it due to lack of research funds to obtain enough amount of the GDNF protein. We hope that the Reviewer understands our limitation. Still, combining the results from New Figure 4H in this study with the New Figure 4F, where GDNF injection induced *Bdnf* mRNA expression in FAPs, and BDNF protein expression in FAPs in response to nerve injury was demonstrated via western blot, we anticipate that GDNF injection would increase BDNF protein levels in FAPs, though direct validation of this statement would require conducting the additional experiments mentioned by the Reviewer.

References

Chan JR, Cosgaya JM, Wu YJ, and Shooter EM (2001). Neurotrophins are key mediators of the myelination program in the peripheral nervous system. *Proceedings of the National Academy of Sciences* 98:14661-14668.

English AW, Liu K, Nicolini JM, Mulligan AM, and Ye K (2013). Small-molecule trkB agonists promote axon regeneration in cut peripheral nerves. *Proc Natl Acad Sci U S A* 110:16217-22.10.1073/pnas.1303646110

Giordani L, He GJ, Negroni E, Sakai H, Law JY, Siu MM, Wan R, Corneau A, Tajbakhsh S, and Cheung TH (2019). High-dimensional single-cell cartography reveals novel skeletal muscle-resident cell populations. *Molecular Cell* 74:609-621. e6.

Höke A, Gordon T, Zochodne D, and Sulaiman O (2002). A decline in glial cell-linederived neurotrophic factor expression is associated with impaired regeneration after long-term Schwann cell denervation. *Experimental neurology* 173:77-85.

Kim J-H, Kang J-S, Yoo K, Jeong J, Park I, Park JH, Rhee J, Jeon S, Jo Y-W, and Hann S-H (2022). Bap1/SMN axis in Dpp4+ skeletal muscle mesenchymal cells regulates the neuromuscular system. *JCI Insight* 7:

Leinroth AP, Mirando AJ, Rouse D, Kobayahsi Y, Tata PR, Rueckert HE, Liao Y, Long JT, Chakkalakal JV, and Hilton MJ (2022). Identification of distinct non-myogenic skeletal-muscle-resident mesenchymal cell populations. *Cell Reports* 39:

Liu L, Cheung TH, Charville GW, and Rando TA (2015). Isolation of skeletal muscle stem cells by fluorescence-activated cell sorting. *Nature protocols* 10:1612-1624.

Oudega M, and Hagg T (1998). Neurotrophins promote regeneration of sensory axons in the adult rat spinal cord. *Brain Research* 818:431-438.10.1016/S0006-8993(98)01314-6

Xiao J, Wong AW, Willingham MM, Kaasinen SK, Hendry IA, Howitt J, Putz U, Barrett GL, Kilpatrick TJ, and Murray SS (2009). BDNF exerts contrasting effects on peripheral myelination of NGF-dependent and BDNF-dependent DRG neurons. *J Neurosci* 29:4016-22.10.1523/JNEUROSCI.3811-08.2009

Xu P, Rosen KM, Hedstrom K, Rey O, Guha S, Hart C, and Corfas G (2013). Nerve injury induces glial cell linederived neurotrophic factor (gdnf) expression in schwann cells through purinergic signaling and the pkcpkd pathway. *Glia* 61:1029-1040.

Zhang JY, Luo XG, Xian CJ, Liu ZH, and Zhou XF (2000). Endogenous BDNF is required for myelination and regeneration of injured sciatic nerve in rodents. *European Journal of Neuroscience* 12:4171-4180.10.1111/j.1460-9568.2000.01312.x

Zheng J, Sun J, Lu X, Zhao P, Li K, and Li L (2016). BDNF promotes the axonal regrowth after sciatic nerve crush through intrinsic neuronal capability upregulation and distal portion protection. *Neuroscience letters* 621:1-8.